# How Well Does GPT-4o Understand Vision? Evaluating Multimodal Foundation Models on Standard Computer Vision Tasks

**Rahul Ramachandran**[‡]          **Ali Garjani**          **Roman Bachmann**[†]

**Andrei Atanov**[*]          **Oğuzhan Fatih Kar**[*,†]          **Amir Zamir**[*]

Swiss Federal Institute of Technology (EPFL)

https://fm-vision-evals.epfl.ch/

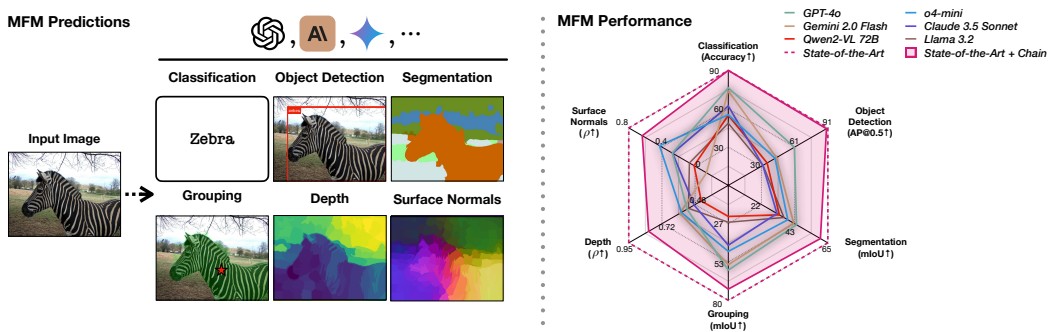

Figure 1: We benchmark multimodal foundation models (MFMs) on established computer vision tasks and datasets. **Left:** GPT-4o's predictions for each task. **Right:** The performance of the MFMs on several classical computer vision tasks. We compare MFMs with specialist models both directly and by calibrating for the chosen structure and constraints of the used prompt chain (+*chain*; see Sec. 4). The axes are normalized using task-specific lower and upper bounds, defined by blind guessing and state-of-the-art specialist performance, respectively.

## ABSTRACT

Multimodal foundation models (MFMs), such as GPT-4o, have recently made remarkable progress. However, their detailed visual understanding beyond question answering remains unclear. In this paper, we **benchmark popular MFMs** (GPT-4o, o4-mini, Gemini 1.5 Pro and Gemini 2.0 Flash, Claude 3.5 Sonnet, Qwen2-VL, Llama 3.2) **on standard computer vision tasks** (semantic segmentation, object detection, image classification, depth and surface normal prediction) **using established datasets** (e.g., COCO, ImageNet and its variants, etc.).

The main challenges in performing this analysis are: **1)** most models are trained to output text and cannot natively express versatile domains, such as segments or 3D geometry, and **2)** many leading models are proprietary and accessible only at an API level, i.e., there is no weight access to adapt them. We address these by translating vision tasks into text-promptable, API-compatible formats via prompt chaining, creating a standardized benchmarking framework.

We observe that: **1)** The MFMs are not close to the state-of-the-art specialist models at any tasks. **2)** They are respectable generalists; this is remarkable as they are presumably trained on primarily image-text-based tasks. **3)** They perform semantic tasks notably better than geometric ones. **4)** GPT-4o performs the best among non-reasoning models, securing the top position in 4 out of 6 tasks. **5)** Reasoning

---

[*]Equal advisorship.

[†]Work done while at EPFL, now at Apple.

[‡]Work done while at EPFL, now at UIUC.

models, e.g. o3, show improvements in geometric tasks. **6)** While prompt chaining techniques affect performance, better models are less sensitive to prompt variations. **7)** An analysis of models with native image generation, such as the latest GPT-4o, shows they exhibit failure modes, such as hallucinated objects or misalignment between input and output.

# 1 INTRODUCTION

Multimodal foundation models (MFMs), such as GPT-4o, Gemini 1.5 Pro and 2.0 Flash, and Claude 3.5 Sonnet (Anthropic, 2024; OpenAI, 2024b; Reid et al., 2024), have advanced rapidly, demonstrating impressive capabilities in their public releases (OpenAI, 2024b). However, while the community has extensively investigated their language proficiency (Chen et al., 2021; Chiang et al., 2024; Hendrycks et al., 2020; Rein et al., 2023), the extent of their vision capabilities remains underexplored. We still lack a well-calibrated quantitative understanding of their performance on established vision tasks and datasets, particularly across diverse axes of vision, e.g. semantics, 3D, grouping, etc.

Most existing vision benchmarks for MFMs primarily target text (e.g., VQA) or tasks closely tied to text, such as classification (Fu et al., 2024; Rahmanzadehgervi et al., 2024; Tong et al., 2024a;b; Wu and Xie, 2024; Yue et al., 2024). While they provide useful insights, several key limitations persist. First, it is unclear how much solving these benchmarks truly depends on the visual input, and some were shown to mainly measure the language capabilities of MFMs while overlooking the vision component (Tong et al., 2024a). Second, they all require the model to output text, making it hard to compare the vision capabilities of MFMs on vision-only tasks against vision specialist models. Third, they do not shed light on other aspects of visual understanding, such as 3D geometry, grouping, or segmentation, that are less text-oriented.

We address these limitations by evaluating MFMs on well-established vision tasks and datasets developed by the community. Specifically, we test GPT-4o, o4-mini, Claude 3.5 Sonnet, Gemini 2.0 Flash, Gemini 1.5 Pro, Qwen2-VL, and Llama 3.2 on classification, object detection, semantic segmentation, grouping, depth prediction, and surface normal prediction using COCO (Lin et al., 2014), Hypersim (Roberts et al., 2021), as well as ImageNet (Russakovsky et al., 2014) and its variants (Hendrycks and Dietterich, 2019; Hendrycks et al., 2021; Kar et al., 2022b; Recht et al., 2019; Wang et al., 2019). Most of these tasks, however, require dense pixel-wise predictions not readily compatible with the default text output of most MFMs. Furthermore, direct prompting usually leads to a varying and often weak performance across tasks, hence it may not represent the actual visual understanding capabilities of MFMs (see Sec. 4.2 and App. E).

To address these challenges, we split each task into multiple sub-tasks, each of which can be solved in a textual form via prompting (see Sec. 3). This results in a *prompt chaining framework that can be applied to any MFM with a text interface (e.g., chatbot APIs) to solve standard vision tasks*. Specifically, our proposed approach allows MFMs to **1)** detect bounding boxes, **2)** generate complete segmentation masks for complex scenes, **3)** extract semantic entities from images similar to SAM (Kirillov et al., 2023b), **4)** estimate dense depth and surface normal maps (see Fig. 1.)

We emphasize that this prompt chaining framework is *not proposed as an alternative methodology for solving vision tasks, nor do we suggest that MFMs should adopt such approaches in practice.* Rather, our framework serves specifically as a standardized method to measure and benchmark *any MFM* that can input images and output text. Crucially, *this enables a quantifiable and holistic understanding of MFMs' vision capabilities on various established vision tasks and benchmarks, as well as a direct comparison with vision-only models.*

We find that the current generation of *MFMs achieves a good performance in most cases* and are respectable as generalists, with GPT-4o scoring the best in 4 out of 6 tasks. However, *they still lag behind task-specific state-of-the-art vision models in all tasks.* In particular, we find that the MFMs perform geometric tasks significantly worse than semantic ones. Furthermore, we perform a detailed prompt sensitivity analysis for each task and find that, while performance varies for different prompts, better models exhibit less sensitivity. To enable further research in this direction and enable the community to benchmark future MFMs on vision tasks, we open-source our evaluation code and prompt chains, along with interactive visualizations available on our project website.

## 2 RELATED WORK

**Advances in MFMs.** There has been remarkable progress in MFMs (Achiam et al., 2023; Alayrac et al., 2022; Anthropic, 2024; Bai et al., 2023; Beyer et al., 2024; Dai et al., 2023; Li et al., 2023a; Liu et al., 2024; OpenAI, 2024a;b; 2025b; Reid et al., 2024; Team, 2024; Team et al., 2023; Wang et al., 2022; 2024) (see (Yin et al., 2023; Zhang et al., 2024) for surveys), leading to strong performance on multimodal tasks like VQA and instruction following. Despite the progress, it is unclear how well these models perform tasks that require dense visual understanding, which is our main focus.

**Benchmarking vision capabilities of MFMs.** Many works investigate the vision capabilities of MFMs via VQA-style benchmarks that combine visual and textual inputs to generate textual outputs (Al-Tahan et al., 2024; Fu et al., 2024; Jiang et al., 2024a; Li et al., 2023b; Liu et al., 2024; Rahmanzadehgervi et al., 2024; Tong et al., 2024a;b; Yue et al., 2024). While these approaches offer valuable insights, they are incompatible with vision-only models, making direct comparisons difficult. In contrast, *we evaluate MFMs on standard vision tasks*, enabling direct comparison with strong vision specialists to track MFMs' progress. Cambrian-1 (Tong et al., 2024a) evaluates MFMs on vision datasets (Brazil et al., 2023; Lin et al., 2014; Zhou et al., 2017) by repurposing dataset annotations into text format. We differ by translating MFM outputs into the annotation format instead, e.g., segmentation maps. To the best of our knowledge, this is the first approach that enables an apples-to-apples comparison with vision specialist models using standard task-specific metrics and qualitative analyses in the tasks' native output space.

**Prompting MFMs.** Various prompting techniques have been developed for MFMs (Khot et al., 2022; Wei et al., 2022; Yao et al., 2024; Zhou et al., 2022). We follow a similar strategy and decompose complex vision tasks into simpler sub-tasks that MFMs can handle. Several works developed prompting techniques to unlock the vision capabilities of MFMs (Hu et al., 2024; Wu and Xie, 2024; Wu et al., 2024; Yang et al., 2023a). The related DetToolChain (Wu et al., 2024) for object detection is not fully reproducible at the time of writing. We differ by **1)** focusing on a wider range of tasks including semantic and geometric ones **2)** for several MFMs including closed- and open-weight ones **3)** with a simpler yet effective and uniform prompt chaining mechanism.

## 3 PROMPT CHAINING FOR SOLVING VISION TASKS

In this section, we describe the developed prompt chaining techniques that enable MFMs to solve standard computer vision tasks. The core idea is to break each task into simpler, text-solvable sub-tasks, e.g., identifying whether an object is present in a patch of an image. We then solve each sub-task by prompting the MFM. To guide the choice of how to split each task into sub-tasks, we rely on our early key observation that most MFMs are relatively strong at image classification (see, e.g., Tab. 1) and, therefore, split each task into multiple classification sub-tasks. We provide the pseudo-code for each technique in App. D.

**Object detection.** The goal is to predict bounding box coordinates that tightly localize the objects in the image. Similar to Yang et al. (2023b), our initial attempts showed that many MFMs fail at predicting the coordinates directly. Thus, we develop a prompt chaining method and divide the original task into two stages. The first stage is to identify all objects in the image, similar to classification. In the second stage, for each object, we regress its coordinates via recursive zooming (see Fig. 2). Specifically, we divide the image into grid cells and ask the model to identify whether (a part of) the object is present in each cell. We then discard cells without objects, reducing the search space. We apply this process recursively, progressively eliminating irrelevant regions of the image until only the object of interest remains present in the image. We use two grid resolutions: a coarse grid for quick downsampling and a finer grid for precise edge refinement to reduce the number of steps.

**Semantic segmentation.** The goal is to assign a single semantic class to each pixel in an image. Instead of per-pixel querying that is prohibitively expensive, we split the image into pixel groups using an *unsupervised* superpixel clustering algorithm (Achanta et al., 2012) and assign a single label per group to decrease the number of API calls (or forward passes). This approach is used primarily for **cost efficiency**, as the superpixel algorithm is **semantically-agnostic**: it only provides candidate regions based on low-level features like color and texture (Stutz et al., 2018), while the MFM remains solely responsible for the semantic labeling. We confirm in App. E.8 that our findings hold even when increasing the superpixel granularity. In all our comparisons, *we account for potential biases*

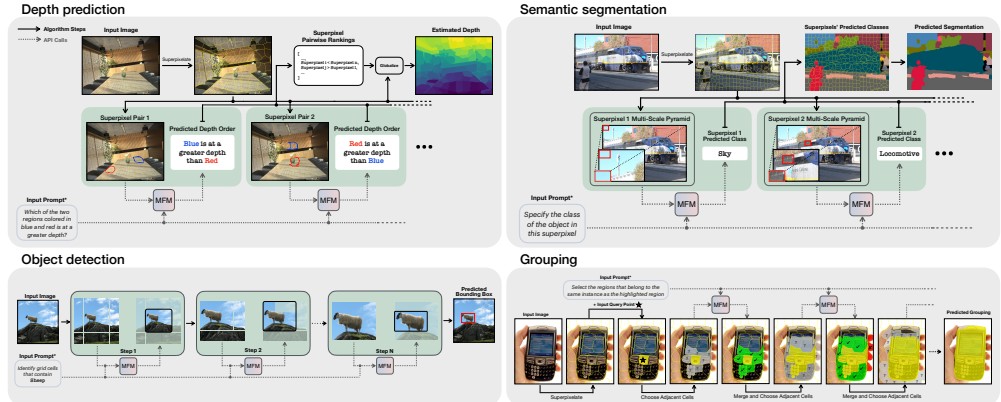

Figure 2: **Prompt chain algorithms overview. a) Depth prediction** randomly samples pairs of superpixels and performs pairwise depth comparisons, which are globalized by minimizing an objective function, resulting in a relative depth map. Surface normal estimation is performed in a similar manner (see App. D.4). **b) Semantic segmentation** constructs multi-scale "pyramids" of superpixel crops (superpixel, local context, full image) and classifies these sequentially with the MFM. **c) Object detection** iteratively queries a $3 \times 3$ grid of image crops for the target class (e.g., Sheep), discarding empty cells and refining the search until the object is localized. **d) Grouping** grows a cluster around a query point by merging adjacent superpixels predicted to belong to the same object. *Summary of the actual prompt; see full prompt chain descriptions in the supplementary material.*

*introduced by superpixelation (and other approximations in prompting)* by introducing calibration control baselines (see Sec. 4.)

After dividing the image into superpixels, we classify them in batches to decrease the overall cost, as in the classification task. Similar to the object detection algorithm, this approach utilizes the strength of MFMs as strong image classifiers. To maintain consistency, we include prior predictions in the prompt history, which improves performance.

In our early experiments, we found that simply outlining individual superpixels on an input image leads to poor performance. This aligns with findings that MFMs have "blurry vision" and struggle with fine-grained localization (Fu et al., 2024; Wu and Xie, 2024). To address this, we provide the MFM with the crops of each superpixel at multiple scales, which we found to improve the performance significantly. Please see Fig. 2 for an overview.

**Grouping.** Given an image and a query (or anchor) point on it, the goal is to identify other pixels that belong to the same object or background. Unlike semantic segmentation, this task has no predefined classes, making it more challenging. To tackle this task, as before, we use superpixels and the MFMs' capability to determine visual similarity (Fu et al., 2024). We create a graph of neighboring superpixels, identify the one with the query point, and explore its neighbors. The model decides whether each adjacent superpixel belongs to the same object as the initial superpixel. The selected superpixels are then merged with the initial one to form the next input cluster. This process continues until no more superpixels are added. Please see Fig. 2 for an overview.

**Depth prediction.** We perform relative depth prediction by querying the model to rank different parts of the image according to their distance from the camera. As per-pixel querying is infeasible, we adopt a region-wise comparison strategy inspired from Zoran et al. (2015). To identify suitable regions for comparison, we first segment the image into superpixels. We then randomly sample pairs of superpixels and query the MFM to rank these pairs based on relative depth (see Fig. 13 in the appendix). These pairwise rankings are then globalized by minimizing the objective function from (Zoran et al., 2015), which assigns larger values to superpixels ranked deeper in pairwise comparisons (see App. D.3). We then use the values found by the optimization method to rank all superpixels. For simplicity, we assume that all pixels within a superpixel share the same depth rank, allowing us to extend the superpixel-level depth predictions to a pixel-wise ranking across the entire image (control baselines are included in evaluations).

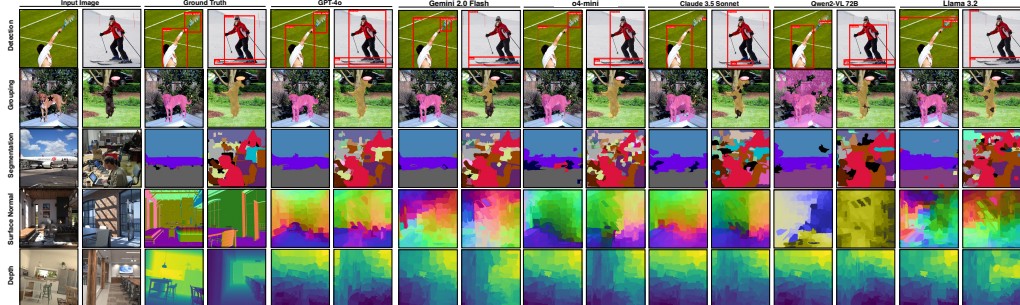

Figure 3: **Qualitative results.** Visual comparisons showing the performance of MFMs across each task. We find that all models perform relatively better on semantic tasks compared to the geometric ones. For surface normal visualizations, we combine the per-axis normalized predictions and project them onto the unit sphere. Please see the appendix for additional qualitative results.

**Surface normal prediction.** We follow a ranking approach similar to that used for depth. We use standard basis vectors relative to the camera (right, up, and forward) as reference directions, and for each randomly sampled pair of superpixels, we query the MFM to determine their relative alignment with each basis vector (see Fig. 14 in the appendix). After we obtain the pairwise comparisons for each direction, we globalize them using the same algorithm used for depth (Zoran et al., 2015). This results in three distinct surface normal maps, one for each basis direction. Similar to depth, we assume uniformity within superpixels and assign the same rank to all pixels within each superpixel group (control baselines are included in the evaluations).

**Image classification.** The MFM classifies the image from a list of predefined classes. For efficiency, we follow (Jiang et al., 2024a) and classify images in batches, which does not significantly harm accuracy.

## 3.1 ACCOUNTING FOR ALGORITHMIC CONSTRAINTS

Using superpixels and the zooming algorithm can impose constraints on MFMs' performance. We address this in two ways. First, we introduce *targeted control baselines* in all our experiments (see Sec. 4) to ensure fair and calibrated comparisons with other vision models. Second, in App. E.8, we demonstrate that employing more fine-grained prompt chains (e.g., using more superpixels) yields consistent conclusions, confirming that our findings are robust to the chosen granularity.

## 4 EXPERIMENTS

We present the experimental results for different tasks and MFMs. First, we describe our setting, including the choice of the datasets and models. Then, we discuss our main results. We provide qualitative examples for all tasks in Fig. 3. Finally, we provide further analysis and ablations in Sec. 4.2. Please see the appendix for additional results.

**Evaluated MFMs.** We evaluate closed-weight MFMs, namely GPT-4o (OpenAI, 2024b), Gemini 2.0 Flash (Google DeepMind, 2024), Gemini 1.5 Pro (Reid et al., 2024), and Claude 3.5 Sonnet (Anthropic, 2024) by querying them via their APIs. We also include Qwen2-VL-72B (Wang et al., 2024) and Llama 3.2 90B (AI, 2024; Meta) as recent open-weight models that yield competitive results with GPT-4o and Claude 3.5 on several benchmarks. In addition, we evaluate recent multimodal reasoning models such as o1, o3, and o4-mini (a lightweight distilled variant) (OpenAI, 2024a; 2025b). We evaluate o4-mini across the full benchmark, and evaluate o1 and o3 on smaller representative subsets due to cost constraints. For each model and task, we select the best prompt on a validation set for final testing.

**Datasets.** We use the following common vision datasets for evaluations:

*Image classification.* We use standard benchmarks including ImageNet (Russakovsky et al., 2014) and ImageNet-v2 (Recht et al., 2019). To test robustness, we use ImageNet-R (Hendrycks et al., 2021), ImageNet-S (Wang et al., 2019), and two corruption benchmarks from RobustBench, namely,

Table 1: **Image classification.** We compare the top-1 acc. (↑) of the MFMs with vision specialists, Model Soups (Wortsman et al., 2022) and OpenCLIP (Cherti et al., 2023). Although their performance falls short of the top specialist models, MFMs, particularly GPT-4o, demonstrate competitive results across a broad range of benchmarks.

Table 2: **Grouping.** Comparison of MFMs with SAM and SAM-2 (Kirillov et al., 2023b; Ravi et al., 2024) on the grouping task.

| Model | ImageNet | ImageNet-V2 | Corruptions | | Domain Shift | |
|---|---|---|---|---|---|---|
| | | | 2DCC | 3DCC | ImageNet-R | ImageNet Sketch |
| Model Soups ViT-G | 90.94 | 84.22 | - | - | 95.46 | 74.23 |
| OpenCLIP H | 84.37 | 78.33 | 66.96 | 65.95 | 93.76 | 73.24 |
| GPT-4o | 77.20 | 71.57 | 62.46 | 61.13 | 84.38 | 67.30 |
| o4-mini | 55.90 | 46.99 | 37.22 | 36.68 | 56.05 | 45.18 |
| Gemini 2.0 Flash | 74.78 | 75.79 | 55.67 | 56.92 | 82.05 | 69.43 |
| Gemini 1.5 Pro | 73.88 | 69.76 | 56.14 | 56.22 | 71.42 | 57.15 |
| Claude 3.5 Sonnet | 62.85 | 54.45 | 40.76 | 41.41 | 70.36 | 57.42 |
| Qwen2-VL | 55.54 | 49.39 | 38.92 | 36.45 | 66.31 | 51.18 |
| Llama 3.2 | 49.15 | 48.21 | 34.45 | 34.37 | 65.05 | 47.11 |

| Model | mIoU (↑) |
|---|---|
| SAM-2 | 80.87 |
| SAM-2 + Chain | 72.35 |
| SAM | 80.12 |
| SAM + Chain | 72.32 |
| GPT-4o | 59.06 |
| o4-mini | 46.00 |
| Gemini 2.0 Flash | 55.25 |
| Gemini 1.5 Pro | 44.13 |
| Claude 3.5 Sonnet | 41.68 |
| Qwen2-VL | 21.64 |
| Llama 3.2 | 25.69 |
| Oracle + Chain | 81.77 |

ImageNet-C (Croce et al., 2020; Hendrycks and Dietterich, 2019) and ImageNet-3DCC (Kar et al., 2022b).

*Object detection.* We use the COCO (Lin et al., 2014) validation set and choose images containing a single instance of each present class, resulting in 1.7K examples.

*Semantic segmentation & grouping.* We use a random subset of 500 COCO (Lin et al., 2014) validation images for semantic segmentation for cost-efficiency. For grouping, we filter 100 images from the COCO validation set by measuring the consistency of SAM (Kirillov et al., 2023a) predictions between different query points within every instance. More details are provided in the appendix.

*Depth & surface normal prediction.* We use Hypersim (Roberts et al., 2021) and randomly subsample 100 validation images from it for cost-efficiency.

**Baselines.** We include the following control baselines to contextualize the performance of MFMs and account for the impact of superpixelation and other design choices:

*Vision Specialist.* We report the performance of leading vision models for each task. We specify each model used in the corresponding task sections. In addition to the state-of-the-art models, we benchmark 4M-21 (Bachmann et al., 2024; Mizrahi et al., 2023) as a zero-shot vision baseline for an unbiased comparison. Although 4M-21 is an MFM, here we treat it as a vision specialist because it is specifically trained for solving these tasks. Overall, these baselines indicate the current state of the (specialized) computer vision models.

*Vision Specialist + Chain.* This control baseline applies the same algorithmic constraints to the vision specialist as those experienced by MFMs, such as superpixels and recursive zooming. This control baseline provides a **fair and calibrated comparison** between vision specialists and MFMs, ensuring that our benchmark remains accurate in a relative and ordinal sense.

*Oracle + Chain.* This baseline shows the performance of the prompt chain if the MFM gave the ground-truth answer for each sub-task. *It isolates MFM performance from limitations imposed by our chain's granularity.* Note that this is not a hard upper bound, and we can alleviate these limitations by using more fine-grained chains (see App. E.8.)

*Blind Guess.* We prompt the model with a blank image, revealing potential biases and assessing whether the model genuinely utilizes the image content for its predictions.

## 4.1 EVALUATION RESULTS

**Object detection.** The results are summarized in Tab. 3. We use DETR (Carion et al., 2020) and Co-DETR (Zong et al., 2023), a state-of-the-art COCO model, as the vision specialists, and 4M-21 as a zero-shot baseline. We observe that all MFMs lag behind the vision models, with GPT-4o achieving the highest performance, significantly outperforming other MFMs. For the "Specialist + Chain" baselines, we apply the same recursive algorithm, but at each stage we only keep the grid cells that intersect with the original bounding boxes predicted by the specialist. As mentioned earlier, this calibration confirms that the gap between MFMs and specialists remains significant.

Table 3: **Object Detection:** Average precision of MFMs vs. vision specialists (DETR, Co-DETR) and 4M-21. GPT-4o leads among MFMs.

| Baselines | Model | AP$_{50}$ (↑) | AP$_{75}$ (↑) | AP (↑) |
|---|---|---|---|---|
| | Co-DETR | 91.30 | 86.17 | 80.23 |
| | Co-DETR + Chain | 90.06 | 52.78 | 51.54 |
| Vision Specialists | DETR | 73.31 | 63.61 | 58.67 |
| | DETR + Chain | 72.33 | 38.36 | 39.36 |
| | 4M-21 | 59.54 | 51.57 | 47.71 |
| | 4M-21 + Chain | 55.46 | 30.48 | 30.74 |
| | GPT-4o | 60.62 | 31.97 | 31.87 |
| | o4-mini | 42.90 | 22.18 | 22.60 |
| | Gemini 2.0 Flash | 44.17 | 15.83 | 19.85 |
| MFMs | Gemini 1.5 Pro | 39.75 | 15.27 | 18.11 |
| | Claude 3.5 Sonnet | 31.69 | 12.13 | 14.78 |
| | Qwen2-VL | 35.62 | 12.82 | 15.27 |
| | Llama 3.2 | 31.87 | 8.40 | 12.83 |
| | Oracle + Chain (pred. class) | 75.44 | 41.31 | 41.56 |
| Control | Oracle + Chain (full) | 92.18 | 49.33 | 50.14 |
| | Blind guess | <0.01 | <0.01 | <0.01 |

Table 4: **Semantic Segmentation:** mIoU and pixel accuracy of MFMs vs. OneFormer and 4M-21. GPT-4o again performs best among MFMs.

| Baselines | Model | mIoU (↑) | Pixel acc. (↑) |
|---|---|---|---|
| | OneFormer | 65.52 | 83.26 |
| Vision Specialists | OneFormer + Chain | 60.64 | 81.69 |
| | 4M-21 | 54.31 | 79.66 |
| | 4M-21 + Chain | 52.72 | 78.59 |
| | GPT-4o | 44.89 | 68.60 |
| | o4-mini | 39.19 | 64.26 |
| | Gemini 2.0 Flash | 43.04 | 66.15 |
| MFMs | Gemini 1.5 Pro | 40.46 | 64.88 |
| | Claude 3.5 Sonnet | 32.05 | 58.41 |
| | Qwen2-VL | 33.59 | 56.36 |
| | Llama 3.2 | 36.63 | 59.95 |
| Control | Oracle + Chain | 83.41 | 94.68 |
| | Blind guess | 0.03 | 0.29 |

Finally, we assess the performance of the "Oracle + Chain" baseline. We evaluated two variants: one using GPT-4o's class predictions, and another using the ground-truth class labels. The first baseline examines the outcome if GPT-4o correctly selects the grid cells at each step of the chain, while the second assumes both correct class predictions and accurate grid cell selection. These provide the upper bounds for both the grid search component and the overall pipeline when using a specific grid resolution to calibrate the performance.

**Semantic segmentation.** Table 4 and Fig. 3 show that MFMs achieve rather non-trivial performance, yet they still lag behind the vision specialist, i.e. OneFormer (Jain et al., 2022). Similar to object detection, we include the baseline of constraining the performance of the vision specialist using the chain algorithm: we assign the majority class prediction to each superpixel and flood-fill the entire superpixel with that class.

**Grouping.** Table 2 shows that MFMs have varying performance on this task, and GPT-4o performs the best, achieving overall good performance as can also be seen in Fig. 3. All MFMs still lag behind the vision specialist SAM (Kirillov et al., 2023a).

**Depth prediction.** The results are summarized in Tab. 5. Alongside standard metrics, we also report **1)** the Spearman correlation coefficient ($\rho$), which serves as a relative metric by measuring the correlation between the ground-truth depth ranking of the pixels and the predicted ranking and **2)** accuracy, which reflects the percentage of correct pairwise depth comparisons. MFMs demonstrate a nontrivial performance and outperform the blind guess baseline. Notably, o4-mini achieves the strongest performance among MFMs, despite trailing behind some models in the semantic tasks shown before. Still, there remains a significant gap compared to vision specialists like Omnidata (Eftekhar et al., 2021; Kar et al., 2022a), Lotus (He et al., 2025), and MoGe-2 (Wang et al., 2025), which is more pronounced compared to the semantic tasks.

Similar to previous tasks, we analyze the performance using the "Oracle + Chain" baseline, which assumes 100% accuracy in all pairwise comparisons, and the "Omnidata + Chain" baseline, which uses depth predictions from Omnidata to perform these pairwise comparisons. Due to the coarse granularity of the evaluation, the two baselines closely match each other. Importantly, we find that MFMs still lag behind these baselines, suggesting that our conclusions hold despite the chosen granularity level (see App. E.8).

**Surface normal prediction.** To assess performance, we employ Spearman's rank correlation coefficient, $\rho_i$, measuring the correlation between ground truth and predicted pixel alignments along each basis direction $i$. Alignment for a pixel is measured as the dot product of the surface normal with the direction $i$.

Tab. 6 demonstrates that most MFMs struggle with this task, with some showing a negative correlation for certain directions, revealing a systematic bias in their understanding of these directions. Notably, o4-mini outperforms all other MFMs, indicating stronger geometric understanding. This extends to other recent reasoning models, with o1 and o3 also showing strong performance on the evaluated subset.

Furthermore, we show in the appendix that the MFMs not directly trained for reasoning improve their performance on the up-down ambiguity resolution with CoT prompting (Wei et al., 2022). Similar to

Table 5: **Depth prediction.** MFMs can coarsely estimate depth, but their gap to vision specialists is larger than in semantic tasks. All MFMs perform similarly, with GPT-4o and o4-mini slightly ahead.

| Baselines | Model | Higher is better ↑ | | | |
|---|---|---|---|---|---|
| | | $\delta_1$ | $\delta_2$ | $\delta_3$ | $\rho$ |
| Vision | Omnidata | 0.768 | 0.867 | 0.911 | 0.95 |
| | Omnidata + Chain | 0.568 | 0.772 | 0.864 | 0.81 |
| | Lotus | 0.776 | 0.861 | 0.913 | 0.93 |
| | Lotus + Chain | 0.578 | 0.779 | 0.866 | 0.82 |
| | Moge | 0.795 | 0.859 | 0.903 | 0.90 |
| | Moge + Chain | 0.576 | 0.774 | 0.861 | 0.82 |
| | 4M-21 | 0.636 | 0.814 | 0.888 | 0.89 |
| | 4M-21 + Chain | 0.565 | 0.774 | 0.865 | 0.81 |
| MFMs | GPT-4o | 0.461 | 0.716 | 0.840 | 0.54 |
| | o4-mini | 0.467 | 0.718 | 0.841 | 0.58 |
| | Gemini 2.0 Flash | 0.461 | 0.715 | 0.839 | 0.59 |
| | Gemini 1.5 Pro | 0.458 | 0.709 | 0.835 | 0.51 |
| | Claude 3.5 Sonnet | 0.428 | 0.693 | 0.830 | 0.49 |
| | Qwen2-VL | 0.432 | 0.698 | 0.831 | 0.44 |
| | Llama 3.2 | 0.458 | 0.711 | 0.835 | 0.53 |
| Control | Oracle + Chain | 0.571 | 0.774 | 0.863 | 0.83 |
| | Blind Guess | 0.375 | 0.628 | 0.773 | 0.25 |

Table 6: **Surface normal prediction.** For surface normal prediction, MFMs again show a large gap to specialists, particularly on the $x$-axis. o4-mini is the strongest MFM, while Gemini often fails, performing near chance.

| Baselines | Model | $\rho_x$ | $\rho_y$ | $\rho_z$ |
|---|---|---|---|---|
| Vision | Omnidata | 0.80 | 0.83 | 0.78 |
| | Omnidata + Chain | 0.64 | 0.70 | 0.58 |
| | Lotus | 0.77 | 0.80 | 0.75 |
| | Lotus + Chain | 0.64 | 0.70 | 0.54 |
| | Moge | 0.80 | 0.80 | 0.77 |
| | Moge + Chain | 0.64 | 0.69 | 0.56 |
| | DSINE | 0.76 | 0.79 | 0.73 |
| | DSINE + Chain | 0.63 | 0.69 | 0.55 |
| | 4M-21 | 0.71 | 0.74 | 0.65 |
| | 4M-21 + Chain | 0.65 | 0.70 | 0.56 |
| MFMs | GPT-4o | -0.14 | 0.57 | 0.40 |
| | o4-mini | 0.22 | 0.61 | 0.46 |
| | Gemini 2.0 Flash | -0.39 | -0.04 | 0.02 |
| | Gemini 1.5 Pro | -0.17 | -0.57 | 0.04 |
| | Claude 3.5 Sonnet | -0.19 | 0.61 | 0.40 |
| | Qwen2-VL | 0.09 | -0.07 | 0.02 |
| | Llama 3.2 | 0.41 | -0.42 | 0.22 |
| Control | Oracle + Chain | 0.64 | 0.70 | 0.60 |
| | Blind guess | -0.48 | -0.61 | 0.11 |

depth, *these results suggest that while MFMs have limited 3D visual understanding, newer reasoning models like o1, o3, and o4-mini exhibit promising progress.*

**Image classification.** The classification results across all datasets are summarized in Tab. 1. We use Model Soups ViT-G (Wortsman et al., 2022) as the vision specialist, and we also include OpenCLIP H (Cherti et al., 2023) to assess zero-shot capabilities. Although MFMs do not reach the performance levels of vision specialists, they demonstrate strong results and resilience to corruptions and distribution shifts. Notably, GPT-4o and Gemini 2.0 Flash stand out with a particularly strong performance, followed by Gemini 1.5 Pro, Claude 3.5 Sonnet, o4-mini, Qwen2-VL, and Llama 3.2. We also observe that o4-mini is especially sensitive to the batch size used during inference, with performance improving at smaller batch sizes (see ablation in the appendix).

**Reasoning models.** As discussed earlier, we evaluate o1 and o3 on a representative subset of images (see App. F for details). For comparison, we also include GPT-4o as a baseline, along with evaluations of o4-mini at varying levels of reasoning effort (*low, medium, and high*). The results are presented in Fig. 4. o1 and o3 slightly outperform GPT-4o on most semantic tasks, and significantly outperform it on geometric tasks. As before, GPT-4o excels in semantic tasks on this subset, while o4-mini performs better on geometric ones. We find no clear trend with the reasoning effort; *medium* and *high* reasoning improve results over *low*, but not always. We refer the reader to the appendix for further details and experiments.

**GPT-4o with image generation capability.** Recent MFMs, such as the updated GPT-4o (OpenAI, 2025a), can generate dense image outputs rather than being restricted solely to output text. This capability represents a promising advancement, potentially enabling a comprehensive evaluation on diverse vision tasks. However, the current image generation capability exhibit several limitations, some of which we illustrate in Fig. 5. Specifically, we observe that generated outputs suffer from spatial misalignments and hallucinations, as also observed in Chen et al. (2025).

Figure 4: **Evaluation of reasoning models.** The performance of o1, o3, and o4-mini (at varying levels of reasoning effort) is compared against GPT-4o on a representative subset of images. The reasoning models exhibit a particularly strong performance in geometric tasks.

This presents challenges for directly applying this model to vision tasks, which we leave to future work. We provide further qualitative examples and preliminary quantitative analyses in App. G.

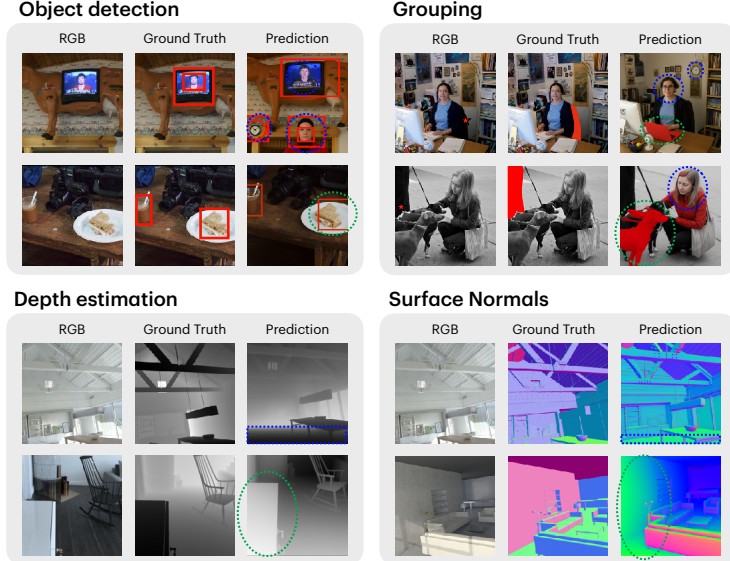

Figure 5: **Failure cases of GPT-4o with image generation capability.** Despite the model's promising capabilities, limitations remain. Here, we highlight some typical failure modes: hallucinations (marked in dotted blue) and inaccurate predictions (marked in dotted green).

## 4.2 ANALYSIS

**Prompt chaining vs direct prompting.** We analyze the impact of using prompt chains versus directly asking the models to solve tasks in a single prompt in Tab. 7. For bounding box regression, we directly query GPT-4o to predict its coordinates; for semantic segmentation, we mark image regions and request corresponding semantic labels. The results indicate a clear performance boost from using the prompt chains. Please see App. E for a detailed discussion, qualitative visuals, and ablations.

**Prompt sensitivity.** The choice of prompt can greatly influence the performance of a model, and we have made reasonable efforts to optimize prompts for each model. We evaluate the MFMs across various prompts to assess their sensitivity to word choice and prompt structure. We then select the most effective prompt on a small validation set for the final results presented in Sec. 4. A comprehensive analysis is provided in App. H, showing that there is some variation in performance with different prompts, and the performance is generally less prompt-dependent for better-performing models, e.g., GPT-4o.

Table 7: **Prompt chaining ablation.** We compare the performance of the prompt chaining algorithm to direct prompting on GPT-4o for semantic segmentation and object detection. *Segmentation is on 100 images.

| Task | Direct prompting | Prompt Chaining (Ours) |
|---|---|---|
| Segmentation (mIoU ↑)* | 25.79 | **41.67** |
| Object Detection (AP$_{50}$ ↑) | 17.69 | **60.62** |

**Analysis of Failure Modes.** While the closed nature of MFMs prevents directly probing internal representations, we observe systematic failure patterns across several axes:

- First, in semantic segmentation, providing full images with the superpixels merely outlined yields poor results (see App. E), whereas multi-scale crops improve performance. This supports findings that MFMs suffer from "blurry vision" and struggle with fine-grained localization (Fu et al., 2024).

- Second, our blind-guess analysis (App. E) reveals a "ceiling bias": models tend to assign larger depth to pixels higher in the image. We observe this trend empirically across several images, where models consistently attribute greater depth to pixels higher in the image.

- Third, in surface normals, models exhibit coordinate-frame ambiguities, frequently inverting the $x$ and $y$-axes (e.g., confusing left and right facing surfaces). We find in App. H that these errors can be partially resolved by CoT prompting.

Overall, MFMs appear to rely heavily on coarse semantic cues while exhibiting unstable fine-grained geometric understanding.

**In-the-wild evaluations.** Previously, we used standard vision datasets like ImageNet and COCO in our evaluations. Given the opaqueness on what training data was used in training the MFMs, one cannot be confident that those images were not used in training. This, so called 'data contamination', problem is a broad concern for the community regarding most MFMs (Jacovi et al., 2023). To assess to what extent the MFMs generalize to entirely novel data and ensure our evaluations are not distorted by potential data contamination, we curated a collection of images released online after the specific model APIs were launched (Flickr, 2024; Unsplash, 2024), which the MFMs could not have encountered before their knowledge cutoff date. Results in App. E.10 show a good generalization performance to the in-the-wild samples. Therefore, we do not find evidence for data contamination with standard datasets to be a concern for our evaluations.

## 5 LIMITATIONS AND CONCLUSIONS

We present a benchmark to investigate the vision capabilities of MFMs by translating standard computer vision tasks into an API-compatible format that can be solved via prompt chaining. Our results show that current MFMs have relatively stronger performance in semantic tasks compared to geometric tasks, and GPT-4o is generally the best-performing model, followed by Gemini 2.0 Flash and 1.5 Pro, o4-mini, Claude 3.5 Sonnet, Llama 3.2, and Qwen2-VL-72B. Despite recent advances, MFMs still lag significantly behind vision specialists, even when specialists are evaluated under the same constraints of the prompt chain. However, recent reasoning models such as o1, o3, and o4-mini show promising performance on geometric tasks, indicating growing capabilities in 3D understanding that complement their already strong semantic performance. We conclude with some limitations and future directions:

*Inference cost.* The multiple API calls per sample (e.g., for classifying batches of superpixels) can result in a higher computational overhead than standard single-query vision-language evaluations (see App. I). While this is a shortcoming, we emphasize that the framework is designed as a **one-time benchmarking tool** for assessing visual capabilities, rather than efficient querying for downstream applications. While efficiency is important, it is orthogonal to our objective, and a promising direction for future work.

*Optimality of the proposed prompt chains.* Research into advanced prompting techniques has the potential to further enhance the performance of MFMs on classical vision tasks, beyond what is shown in this paper. However, as demonstrated by our prompt sensitivity analysis, stronger models generally exhibit a lower reliance on specific prompting strategies, becoming less sensitive to exact prompt formulations. Nonetheless, while the final design appears simple, our proposed prompt chains are carefully designed, emerge from a vast search space, and consistently improve upon direct prompting (Sec. 4.2 and App. E). Furthermore, we also included careful controls and analyses to disentangle the impact of the prompting method from the benchmarking conclusions.

We emphasize that our framework is for benchmarking, not a production-ready method for solving these tasks. Indeed, we expect that future MFMs with any-to-any capabilities will likely close the gap to specialist vision models by training directly on these tasks. Our framework, however, can benchmark any MFM with image-input and text-output capabilities. Through that, we establish the first benchmark for comparing a diverse range of MFMs, both against each other and against specialist vision models, on standard vision tasks.

*Generalization of the Framework.* We view this benchmark as a building block for systematically evaluating image-based semantic and geometric understanding. While our prompt chaining framework is generic, specific tasks may require different discretization strategies (e.g., sliding windows) rather than superpixel grouping. More broadly, extending this method to encompass video understanding and complex visual reasoning is an exciting direction for future work.

### ACKNOWLEDGMENTS

This work was supported as part of the Swiss AI initiative by a grant from the Swiss National Supercomputing Centre (CSCS) under project ID **43** on Alps. This work has received funding from the Swiss State Secretariat for Education, Research and Innovation (SERI). We also acknowledge a gift from Google.

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

# APPENDIX

## TABLE OF CONTENTS

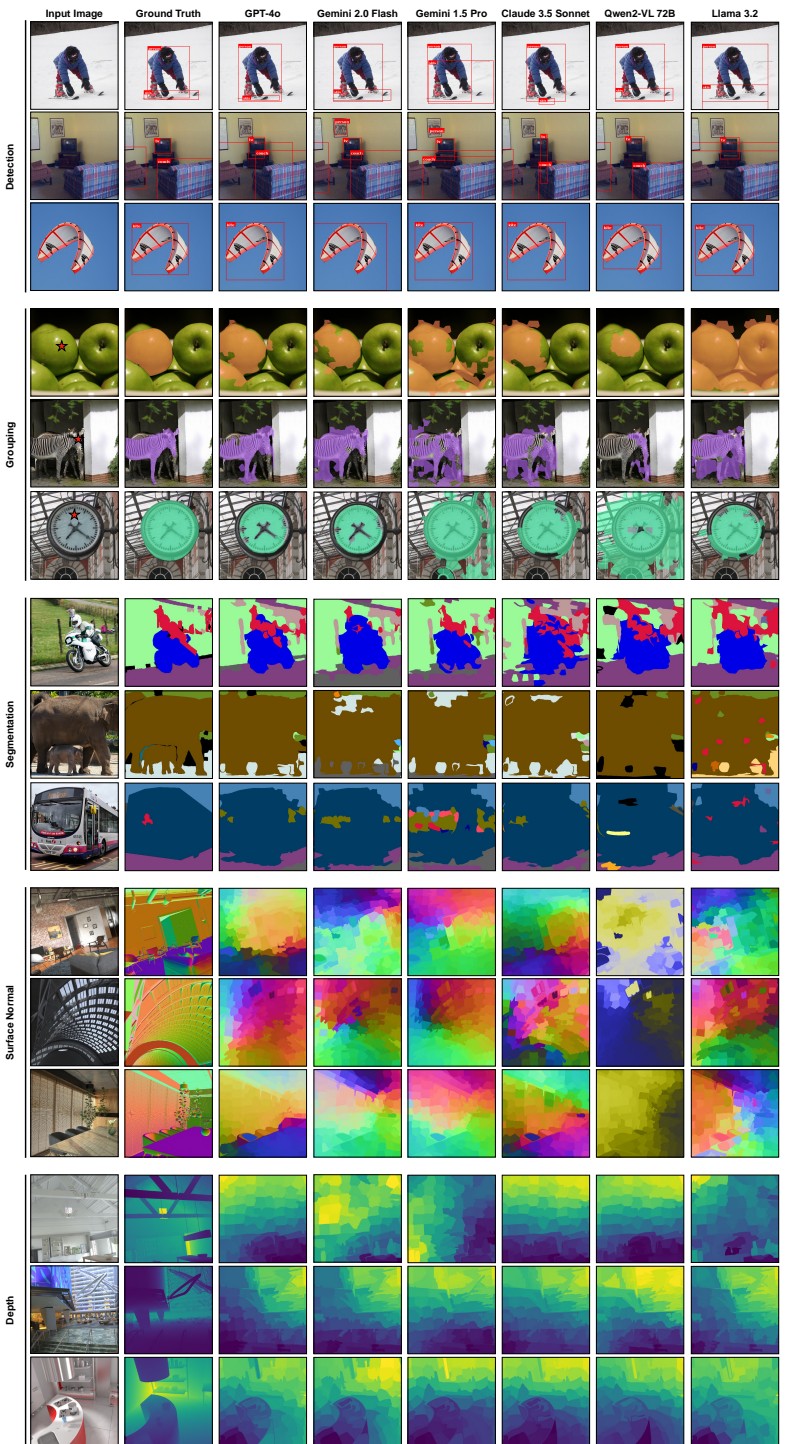

Figure 6: Additional qualitative results for MFM predictions on different tasks.

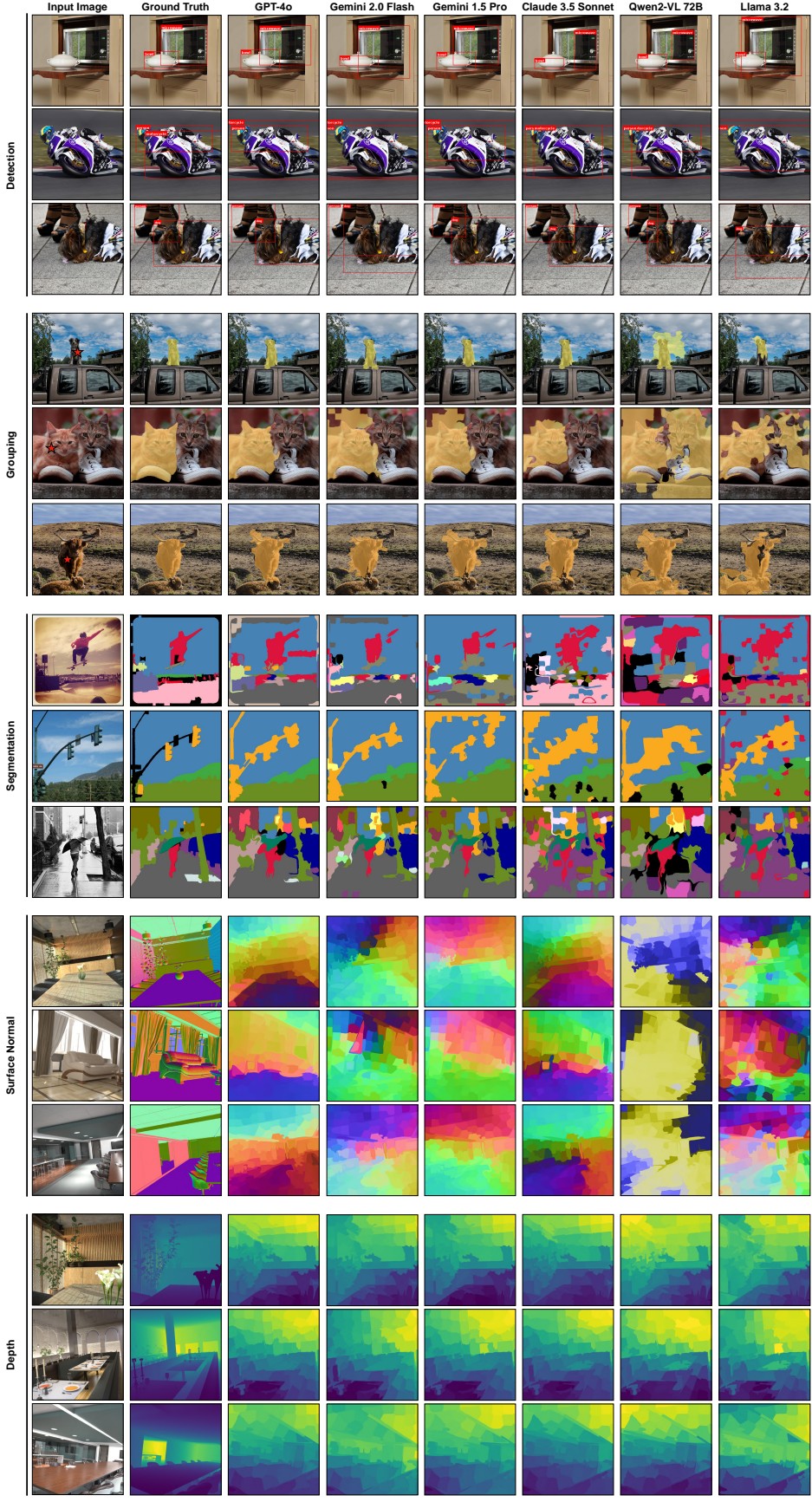

Figure 7: Additional qualitative results for MFM predictions on different tasks.

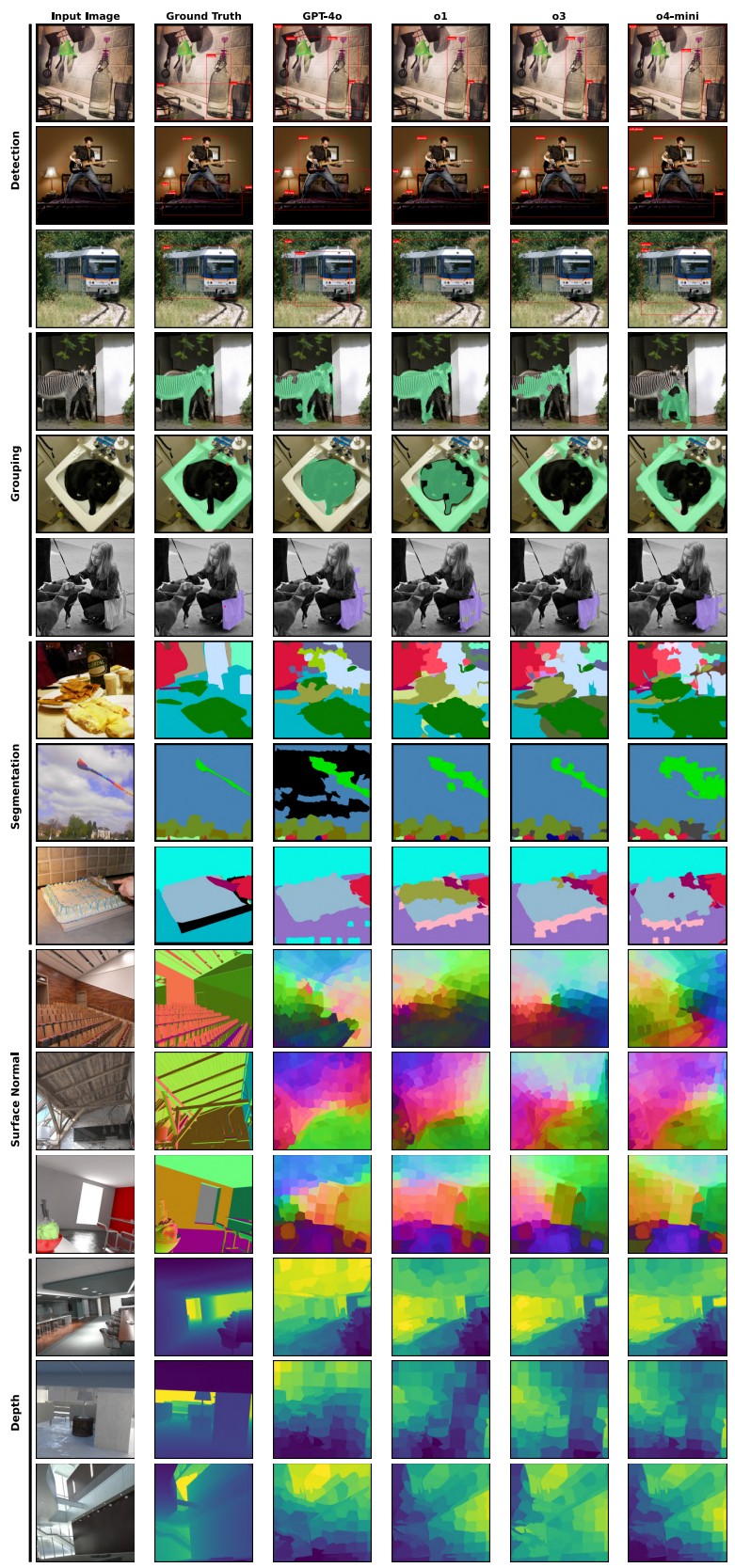

Figure 8: Qualitative results for reasoning model predictions on different tasks next to GPT-4o prediction.

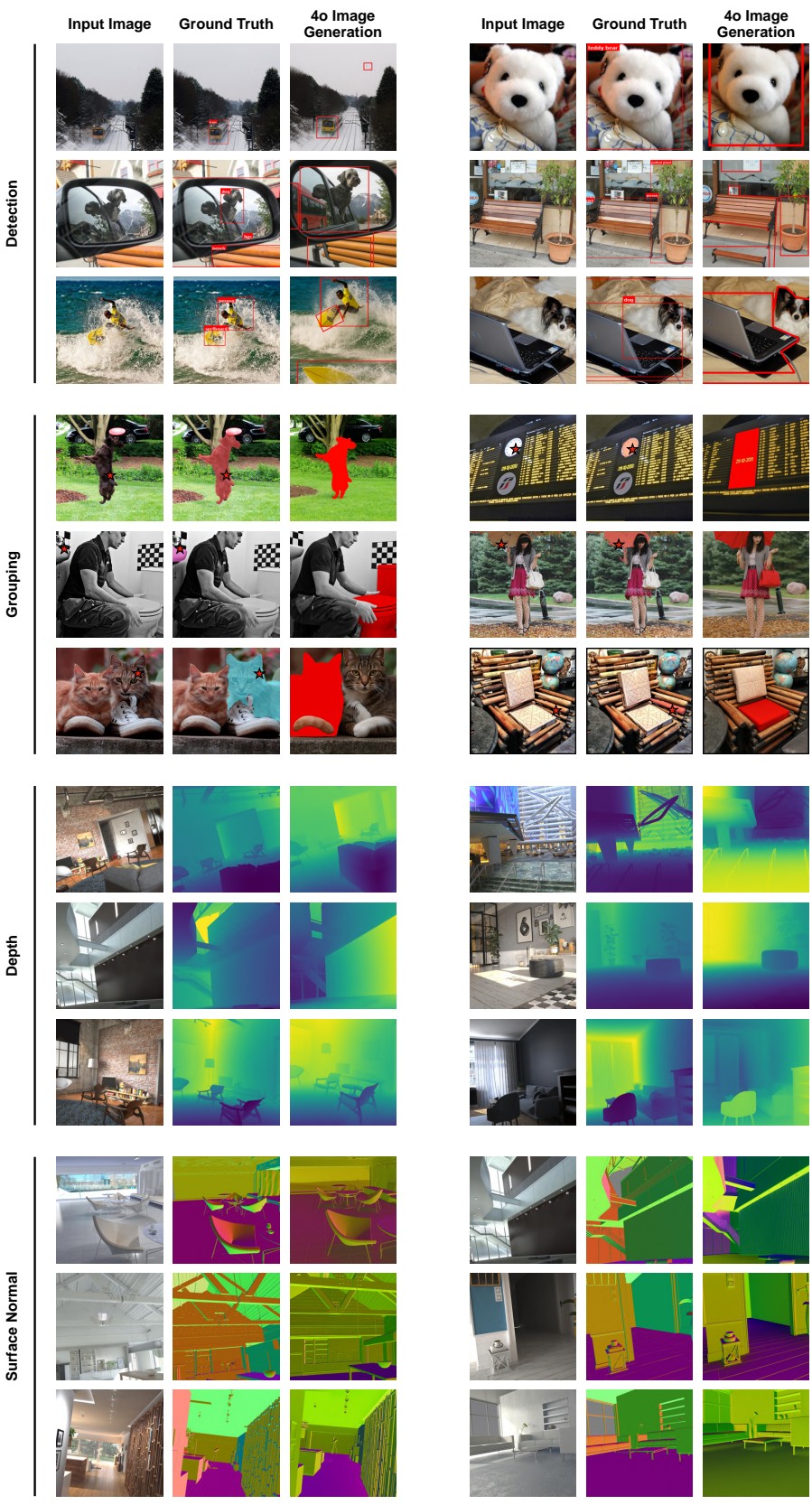

Figure 9: Additional qualitative results for GPT-4o image generation predictions across tasks. Notice the various failure cases such as spatial misalignment in these examples, as outlined in the main paper Fig. 5.

## A    OVERVIEW VIDEO

A narrated overview video including the paper's method, quantitative and qualitative results, as well as interactive visualizations, is provided in the supplementary material.

## B    CODE & FULL PROMPTS

We provide our code and full prompts in the supplementary material.

## C    QUALITATIVE EXAMPLES

We provide additional qualitatives in Figures 6, 7, 8 and 9 to show each model's performance on different tasks.

## D    ADDITIONAL DETAILS ON PROMPT CHAINING

### D.1    OBJECT DETECTION

**Different variations of classification for object detection.** As discussed in Section 3, the first stage of the object detection pipeline involves identifying all the objects present in the image. We attempt the following two strategies for the multi-label classification task:

- The first strategy simply provides the model with the entire image, asking it to identify all present classes.
- The second strategy divides the image into five regions: four quadrants and a center crop. The model is asked to identify the classes present in the 5 regions in independent queries. With each query, the full image is provided for additional context. The final prediction is obtained by taking the union of the classes identified across all regions (see Algorithm 1 in the appendix for detailed pseudocode). This approach typically improves recall but may reduce precision, reflecting a trade-off between the two strategies.

The precision-recall trade-off for the models is described in Tab. 8. To pick the best classification strategy for the models, we run the oracle on the predicted labels on a small subset and pick the one that yields the highest AP.

After we find the object labels, we run the procedure described in Algorithm 2 to regress the bounding boxes. Figure 10 also provides a visualization of the mentioned algorithm.

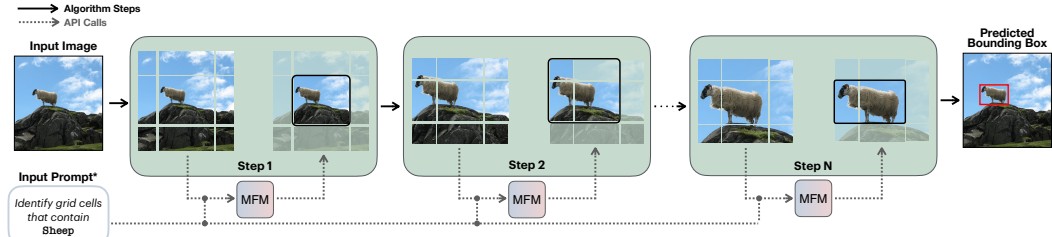

Figure 10: **Object detection algorithm.** At each step, we divide the image into a $3 \times 3$ grid of crops and query each for the presence of the target object (`Sheep` in the figure) through the model. Grid cells without the object are discarded, and the process is repeated until the object is fully located. *Summary of the actual prompt; see the full prompt in the provided Markdown files.*

In Alg. 2, we explored batching the grid-cells and querying them independently. While batching didn't affect results for most MFMs, it significantly deteriorated the performance for Gemini, so we opted to use independent queries for Gemini instead.

Table 8: **Classification for Object Detection:** The results clearly show the precision-recall trade-off between using the two strategies for multi-label classification.

| Strategy | Model | Precision | Recall |
|---|---|---|---|
| Strategy 1 | GPT-4o | 97.5 | 75.75 |
| | Gemini 1.5 Pro | 90.5 | 83.81 |
| | Claude 3.5 Sonnet | 84.27 | 81.24 |
| Strategy 2 | GPT-4o | 89.05 | 88.37 |
| | Gemini 1.5 Pro | 84.37 | 89.3 |
| | Claude 3.5 Sonnet | 78.18 | 85.94 |

---

**Algorithm 1** Region-based Image Classification

---

1: **procedure** REGIONBASEDCLASSIFICATION($image$)
2: $\quad regions \leftarrow$ DivideIntoRegions($image$)
3: $\quad allClasses \leftarrow \emptyset$
4: $\quad$ **for** $region \in regions$ **do**
5: $\quad\quad classes \leftarrow$ QueryMFM($image, region$)
6: $\quad\quad allClasses \leftarrow allClasses \cup classes$
7: $\quad$ **end for**
8: $\quad$ **return** $allClasses$
9: **end procedure**
10: **procedure** DIVIDEINTOREGIONS($image$)
11: $\quad quadrants \leftarrow$ DivideIntoQuadrants($image$)
12: $\quad center \leftarrow$ ExtractCenterCrop($image$)
13: $\quad$ **return** $quadrants \cup \{center\}$
14: **end procedure**

---

**Algorithm 2** Recursive Grid-Search

---

1: **procedure** COARSEGRIDSEARCH($image, object, gridStructure$)
2: $\quad$ **while** search space can be reduced **do**
3: $\quad\quad cells \leftarrow$ DivideIntoGrid($image, gridStructure$)
4: $\quad\quad relevantCells \leftarrow \{c \in cells :$
5: $\quad\quad\quad$ QueryMFM($c, object$) = TRUE$\}$
6: $\quad\quad image \leftarrow$ CropToRelevantCells($image,$
7: $\quad\quad\quad\quad relevantCells$)
7: $\quad$ **end while**
8: $\quad$ **return** $image$ as $bbox$
9: **end procedure**
10: **procedure** QUERYMFM($cell, object$)
11: $\quad$ **return** MFM classification of object presence in cell
12: **end procedure**

---

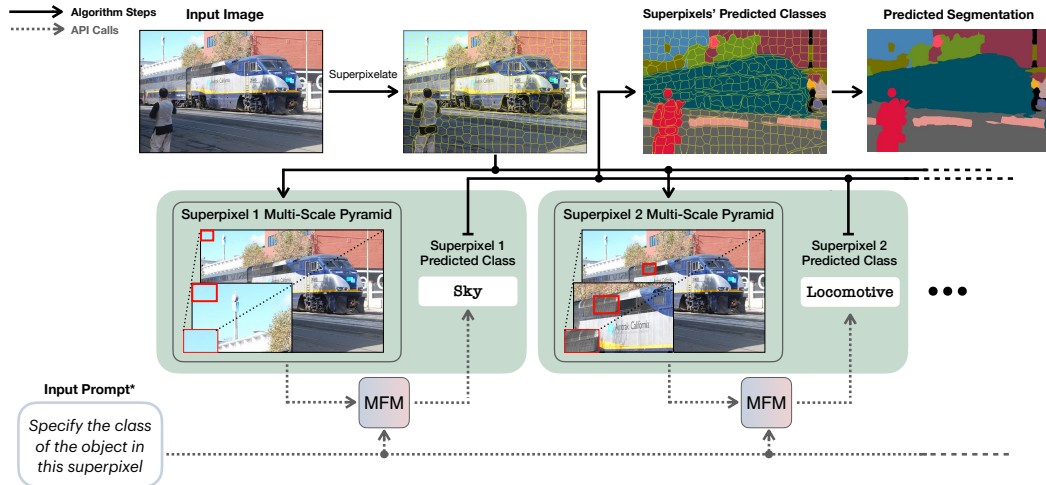

Figure 11: **Semantic segmentation algorithm.** We divide the image into superpixels and create "multi-scale pyramids" of superpixels. The pyramids are then classified using the MFM in a sequential manner to produce the complete segmentation map. A multi-scale pyramid consists of 3 layers: a crop of the superpixel, some context surrounding the crop, and the full image. In practice, we batch multiple superpixels into sequences and classify them jointly. *Summary of the actual prompt; see the full prompt in the provided Markdown files.*

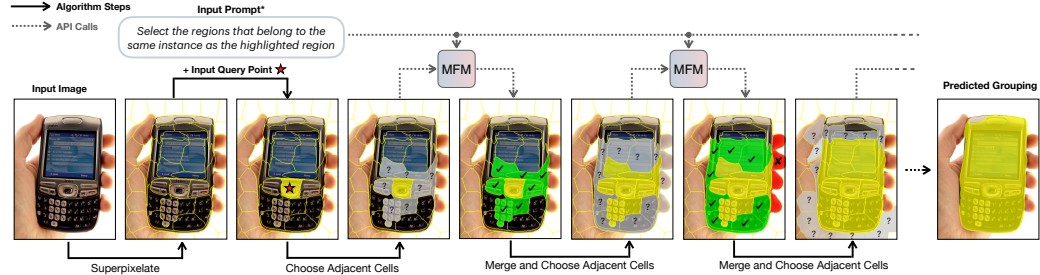

Figure 12: **Grouping algorithm.** Given an image and a query point, we first divide the image into superpixels and select the superpixel that the query point falls into. At each step, the model is asked to identify the adjacent superpixels that belong to the same object as the one covered by the cluster. The selected superpixels are then merged with the cluster to form the next step's input cluster. *Summary of the actual prompt; see the full prompt in the provided Markdown files.*

## D.2 SEGMENTATION

The procedures for supervised segmentation and grouping are described in Algorithm 3 (see Fig. 11) and Algorithm 4 (see Fig. 12) respectively.

## D.3 DEPTH PREDICTION

The procedure for depth prediction is given in Algorithm 5. Furthermore, a visualization of the depth prediction algorithm is given in Fig. 13. A crucial part of the algorithm involves optimizing the objective to obtain the overall depth rankings. To formulate the objective for globalizing the pairwise depth rankings, we repurpose the objective in Zoran et al. (2015). Given the vector of global rankings $\boldsymbol{x} \in \mathbb{R}^N$, we first consider instances where superpixel $i$ is predicted to be at a greater depth than superpixel $j$. The corresponding objective is formulated as:

$$\mathcal{L}_{gt}(\boldsymbol{x}) = \sum_{i,j} (x_i - x_j - 1)^2 \tag{1}$$

---

**Algorithm 3** Superpixel Segmentation

---

1: **procedure** SEMSEGMENTATION($image$, $batchSize$, $scaleList$)
2:  $superpixels \leftarrow$ SLIC($image$)
3:  $classifiedSuperpixels \leftarrow \emptyset$
4:  $history \leftarrow \emptyset$
5:  **for** $i \leftarrow 1$ to length($superpixels$) step $batchSize$ **do**
6:    $batch \leftarrow$ GetBatch($superpixels, i, batchSize$)
7:    $pyramid \leftarrow$ CreateSemanticPyramid(
           $image, batch, scaleList$)
8:    $batchClasses \leftarrow$ ClassifyBatch(
           $pyramid, history$)
9:    $classifiedSuperpixels \leftarrow$
           $classifiedSuperpixels \cup batchClasses$
10:   $history \leftarrow$ UpdateHistory(
           $history, batchClasses$)
11:  **end for**
12:  $segmentedImage \leftarrow$ FloodFillSuperpixels(
           $image, classifiedSuperpixels$)
13:  **return** $segmentedImage$
14: **end procedure**

---

**Algorithm 4** BFS Segmentation

---

1: **procedure** INSTANCEGROUPING($image, queryPoint, batchSize, scaleList$)
2:  $superpixels \leftarrow$ SLIC($image$)
3:  $graph \leftarrow$ ConstructSuperpixelGraph($superpixels$)
4:  $startNode \leftarrow$ FindSuperpixelContaining(
           $superpixels, queryPoint$)
5:  $cluster \leftarrow \{startNode\}$
6:  $queue \leftarrow$ new Queue()
7:  $queue$.enqueue($startNode$)
8:  $visited \leftarrow \{startNode\}$
9:  **while** not $queue$.isEmpty() **do**
10:   $batch \leftarrow$ GetBatchFromQueue($queue, batchSize$)
11:   $batchPyr \leftarrow$ CreateSemanticPyramid(
           $image, batch, scaleList$)
12:   $clusterPyr \leftarrow$ CreateSemanticPyramid(
           $image, cluster, scaleList$)
13:   $newMembers \leftarrow$ QueryMFM(
           $batchPyr, clusterPyr$)
14:   $cluster \leftarrow cluster \cup newMembers$
15:   $queue, visited \leftarrow$ UpdateQueueAndVisited(
           $graph, newMembers, visited$)
16:  **end while**
17:  **return** $cluster$
18: **end procedure**

---

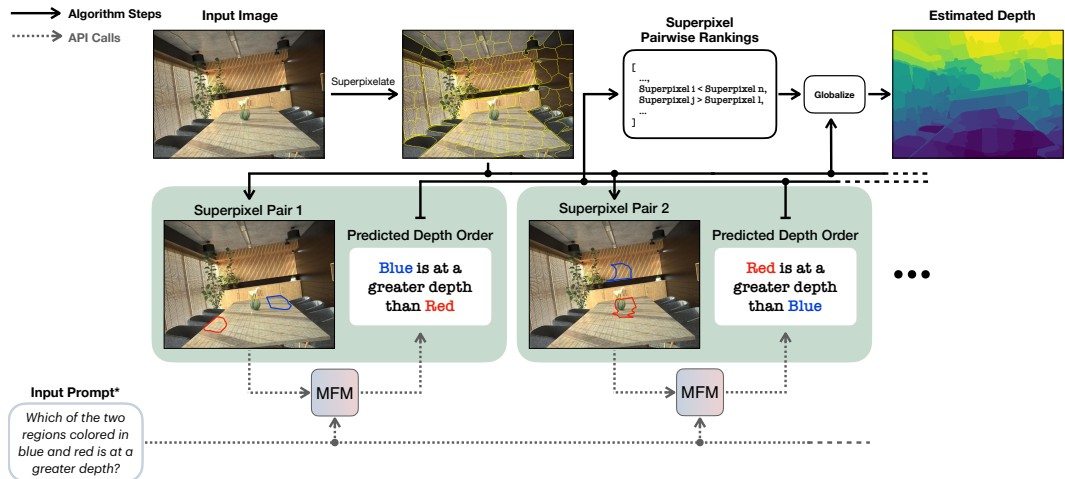

Figure 13: **Depth prediction algorithm**. We randomly select pairs of superpixels. Each pair is given to the model to perform a pairwise depth comparison. The resulting pairwise ranks are then globalized by minimizing an objective function to generate a relative depth map, which can then be scaled to obtain classical evaluation metrics. *Summary of the actual prompt; see the full prompt in the provided Markdown files.*

This objective encourages $x_i$, ranked at a greater depth than $x_j$, to take on higher values. Similarly, an analogous objective $\mathcal{L}_{lt}$ can be defined for superpixels $x_i$ predicted to be at a depth less than $x_j$.

Following Zoran et al. (2015), we include a smoothness regularization term to stabilize the depth estimates:

$$\mathcal{L}_s(\boldsymbol{x}) = \sum_{i,j} (x_i - x_j)^2 \tag{2}$$

This regularization is applied over pairs of adjacent superpixels $i$ and $j$, promoting smooth transitions between their depth values.

The final objective that needs to be minimized is a weighted sum of the above terms:

$$\boldsymbol{x} = \min_{\boldsymbol{x}} \left( \lambda_{gt}\mathcal{L}_{gt} + \lambda_{lt}\mathcal{L}_{lt} + \lambda_s\mathcal{L}_s \right) \tag{3}$$

where $\lambda_{gt}$, $\lambda_{lt}$, and $\lambda_s$ are the weight parameters. For our experiments, we select $\lambda_{gt} = \lambda_{lt} = 1$ and $\lambda_s$ based on the performance on a smaller validation set.

To obtain metric depth estimates, we assume access to ground-truth depth values for the purpose of scaling. Specifically, after floodfilling the values of $\boldsymbol{x}$, we generate a complete relative depth map $\boldsymbol{d}$. Given the ground-truth depth map $\boldsymbol{d}^*$, we optimize the following objective to determine the appropriate scale and shift parameters:

$$(s, t) = \arg\min_{s,t} \sum_{i=1}^{M} (s\boldsymbol{d}_i + t - \boldsymbol{d}_i^*)^2 \tag{4}$$

where $M$ is the total number of pixels in the image. By solving this optimization problem, we can then scale and shift the relative depth map $\boldsymbol{d}$ to align it with the metric depth.

## D.4 SURFACE NORMAL PREDICTION

The procedure for surface normal prediction is detailed in Algorithm 6 and Fig. 14. While the model makes binary decisions regarding whether one depth is lesser or greater than another, we have found that enabling the model to also consider equality predictions enhances the accuracy of surface normal estimates.

---

**Algorithm 5** Depth Prediction

---

1: **procedure** ESTIMATEDEPTH($image, numPairs$)
2:     $superpixels \leftarrow \text{SLIC}(image)$
3:     $pairwiseRankings \leftarrow \emptyset$
4:     **for** $i \leftarrow 1$ to $numPairs$ **do**
5:         $pair \leftarrow \text{SampleRandomPair}(superpixels)$
6:         $ranking \leftarrow \text{QueryMFM}(pair)$
7:         $pairwiseRankings \leftarrow$
                $pairwiseRankings \cup \{ranking\}$
8:     **end for**
9:     $globalRankings \leftarrow \text{MinimizeObjective}($
            $pairwiseRankings)$
10:     $depthMap \leftarrow \text{AssignDepthToPixels}($
            $image, superpixels, globalRankings)$
11:     **return** $depthMap$
12: **end procedure**

---

To incorporate this into our approach, we introduce the following term for cases where superpixels $x_i$ and $x_j$ are predicted to be at equal depth:

$$\mathcal{L}_{eq}(\boldsymbol{x}) = \sum_{i,j}(x_i - x_j)^2 \tag{5}$$

for pairs of superpixels $x_i$ and $x_j$ predicted to lie at an equal depth. For weights, we choose $\lambda_{eq} = \lambda_{lt} = \lambda_{gt} = 1$ and as before, we pick $\lambda_s$ based on the performance on a smaller validation set.

---

**Algorithm 6** Surface Normal Prediction

---

1: **procedure** ESTIMATENORMALS($image, numPairs, bases$)
2:     $superpixels \leftarrow \text{SLIC}(image)$
3:     $pairwiseAlign \leftarrow \{\}$
4:     **for** $i \leftarrow 1$ to $numPairs$ **do**
5:         $pair \leftarrow \text{SampleRandomPair}(superpixels)$
6:         **for** $b$ in $bases$ **do**
7:             $alignment \leftarrow \text{QueryMFM}(pair, b)$
8:             $pairwiseAlign[b] \leftarrow$
                $pairwiseAlign[b] \cup \{alignment\}$
9:         **end for**
10:     **end for**
11:     $normalMaps \leftarrow \{\}$
12:     **for** $b$ in $bases$ **do**
13:         $globalAlign \leftarrow \text{MinimizeGlobalObjective}($
                $pairwiseAlign[b])$
14:         $normalMaps[b] \leftarrow \text{AssignAlignmentToPix}($
                $image, superpixels, globalAlign)$
15:     **end for**
16:     **return** $normalMaps$
17: **end procedure**

---

To visualize surface normals, we take the per-axis predictions and normalize them to [0,1], after which we project them onto the unit sphere. We directly interpret the three channels as RGB values. Note that since the per-axis normalized surface normal predictions do not present absolute directional information with respect to the camera, the colors might not match the ground truth visualizations.

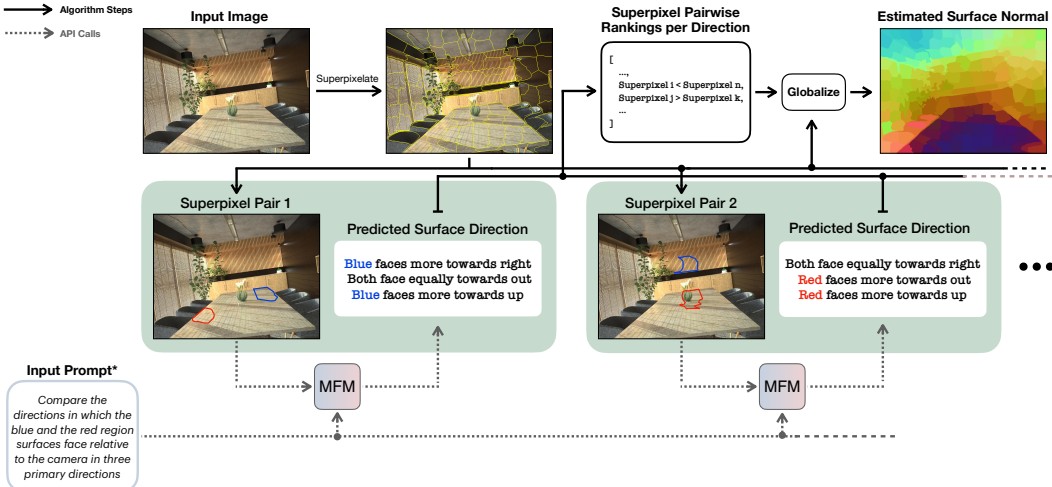

Figure 14: **Surface normal prediction algorithm.** Similar to depth in Fig. 13, we randomly select superpixels and give them to the model to perform a pairwise comparison. The superpixels are compared based on their alignment with the basis vectors relative to the camera. The pairwise ranks are globalized to obtain the final result. *Summary of the actual prompt; see the full prompt in the provided Markdown files.*

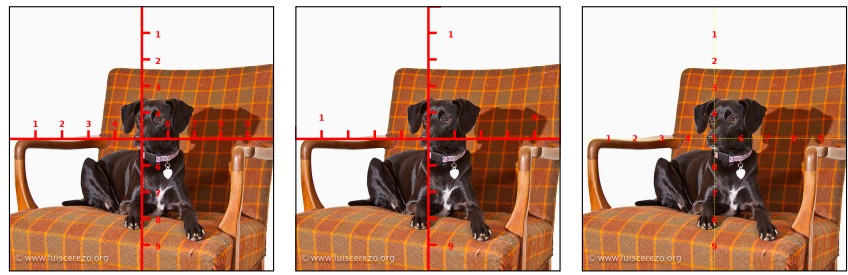

Figure 15: We ablate different ruler types as visual aids for object detection.

## E ADDITIONAL EXPERIMENTAL DETAILS AND RESULTS

### E.1 OBJECT DETECTION

We evaluate additional baselines for GPT-4o in Tab. 9. In these experiments, the classification component of the pipeline remains unchanged, while the grid search is replaced with alternative methods. The results are clear: GPT-4o struggles with directly regressing bounding box coordinates. To address this, we experimented with overlaying rulers on the images to assist in bounding box regression, following insights from Wu et al. (2024), but we found minimal improvement. The various visual prompts we tried are displayed in Fig. 15, and the numbers we obtained on a subset of 100 COCO images are summarised in Tab. 10.

Additionally, we evaluate direct bounding box regression with Gemini, Qwen2-VL, and Claude (see Tab.9), as some of these models have demonstrated this capability (Google, 2024). The results indicate substantial variance in performance: while Gemini and Qwen2-VL localize bounding boxes effectively, GPT-4o and Claude underperform considerably. Interestingly, despite improvements in Gemini and Qwen2-VL, they still lag behind the specialist models and do not surpass GPT-4o when using the chain algorithm.

### E.2 SEMANTIC SEGMENTATION

We depict various marker types used for segmentation in Fig. 16. Furthermore, we conduct an ablation study on the marker type and the context provided during classification, as shown in Tab. 11. The

Table 9: **Additional experiments with MFMs on object detection.** Direct bounding box regression is ineffective for GPT-4o and Claude 3.5 Sonnet, while Gemini 1.5 Pro and Qwen2-VL perform better. For all models, the most effective prompt was selected from a set of options based on validation set performance, similar to the approach used for prompt chains.

| Method | $AP_{50}$ | $AP_{75}$ | AP |
|---|---|---|---|
| GPT-4o (Direct Regression) | 17.69 | 1.69 | 5.08 |
| Gemini 2.0 Flash (Direct Regression) | 38.77 | 10.80 | 15.66 |
| Gemini 1.5 Pro (Direct Regression) | 55.11 | 31.23 | 31.33 |
| Claude 3.5 Sonnet (Direct Regression) | 17.97 | 2.13 | 6.03 |
| Qwen2-VL (Direct Regression) | 44.10 | 23.71 | 24.36 |
| GPT-4o (Regression with Ruler) | 15.95 | 2.60 | 4.99 |

Table 10: **Rulers for Object Detection:** The results indicate that visual markers such as rulers are ineffective in aiding GPT-4o for bounding box regression. Numbers obtained are on a subset of 100 COCO Images.

| Visual Prompt | $AP_{50}$ | $AP_{75}$ | AP |
|---|---|---|---|
| Ruler 1 | 21.19 | 4.09 | 7.60 |
| Ruler 2 | 22.59 | 7.85 | 9.20 |
| Ruler 3 | 19.06 | 4.86 | 8.09 |

**Point marker** **Rectangle marker** **Curve marker**

Figure 16: The curve, rectangle, and point marker types we ablated for the segmentation task.

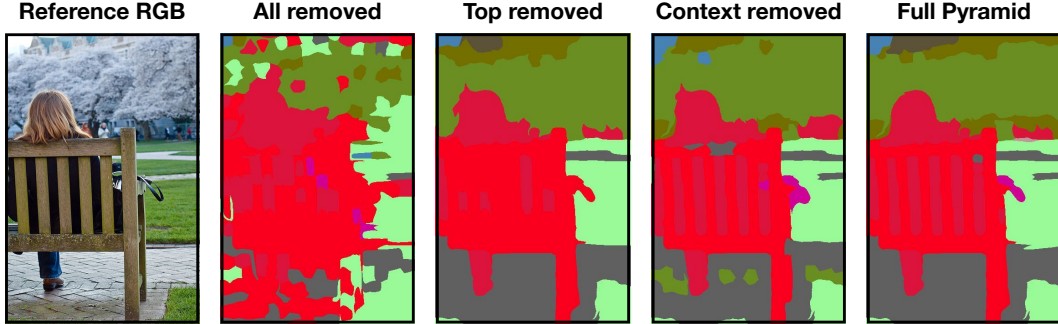

Figure 17: **Semantic Segmentation predictions with different layers of the semantic pyramid**. From left to right: **1.** The RGB Image. **2.** The predicted mask when no crops are given, and markings on the full image are directly used. The model is unable to make out fine details. **3.** The predicted mask when the top of the semantic pyramid is removed. The model misses out on predicting some finer details (for instance, the gaps in the bench and the handbag). **4.** The predicted mask when the middle layer (the context) is removed. The model makes some wrong predictions. **5.** The mask with the full pyramid of information.

**Number markers**     **Rectangle markers**     **Curve markers**

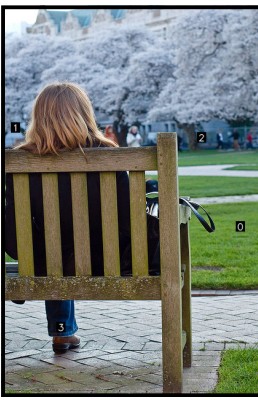 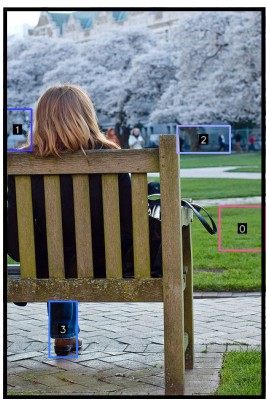 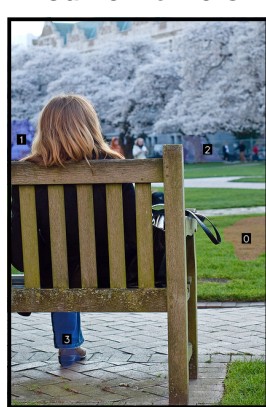

Figure 18: The marker styles used for directly querying semantic entities from the full image.

Table 11: **Ablation study on semantic segmentation.** The results show that GPT-4o is robust to the choice of visual prompt. The substantial performance drop (16 mIoU) observed upon removal of the semantic pyramid shows the critical role of the contextual information used in the sub-task.

| Category | Ablation | mIoU | Pixel Accuracy |
|---|---|---|---|
| Visual Prompts | Curve | 41.62 | 67.43 |
| | Rectangle | 41.67 | 69.74 |
| | Point | 41.68 | 68.83 |
| Contextual Ablations | Without Crop | 40.97 | 69.84 |
| | Without Context | 31.25 | 61.66 |
| | Best Direct | 25.79 | 55.42 |

numbers highlight the importance of contextual information within the semantic pyramid. Removing the context layer leads to a performance drop of over 10 mIoU. Additionally, the direct strategy of marking directly on the image and then classifying results in a 16 mIoU difference, indicating that MFMs currently lack the ability to localize precisely. We also investigate the impact of omitting the finest level of the semantic pyramid—the crop. While the mIoU value does not decrease much, qualitative analysis reveals that this omission hampers the model's ability to capture finer image details. This is shown in Fig. 17.

We also conduct ablation studies on the effect of the model's performance when the semantic pyramid is omitted. The visual markers in Fig. 16 do not work well and do not allow batching, so we borrow a visual marker similar to the one used in Yang et al. (2023a) (see Fig. 18). Table 12 shows the results for different marker types (Yang et al., 2023a). It is clear that the model's performance greatly drops when it is deprived of the crops. We note that the marks we use differ from the ones used in Yang et al. (2023a) in two ways:

- The marks obtained in Yang et al. (2023a) already correspond to semantic entities, while we use superpixels as a proxy for this.

- Extracting a full semantic mask requires discerning finer-grained details, so the marks we use typically correspond to smaller regions in the image.

### E.3 GROUPING

For the grouping task, we filter out 100 COCO images that contain instances which are well-posed for this task. The well-posedness of an instance for grouping is measured by how consistent the SAM predictions are for the instance. To calculate the consistency of predictions for an instance, we sample random points inside the instance and use SAM to obtain an instance mask for each point individually, as well as a global mask by querying all points together. The mIoU between individual masks and

Table 12: **Ablation on Direct Segmentation:** The numbers clearly show that omitting the extra information provided by the crops greatly impacts the model's performance. The numbers shown are for a subset of 30 images. The prompt was selected from a set of options based on validation set performance.

| Visual Marker | mIoU | Pixel Accuracy |
|---|---|---|
| Curve | 20.70 | 50.34 |
| Rectangle | 18.24 | 47.80 |
| Number | 21.13 | 50.00 |

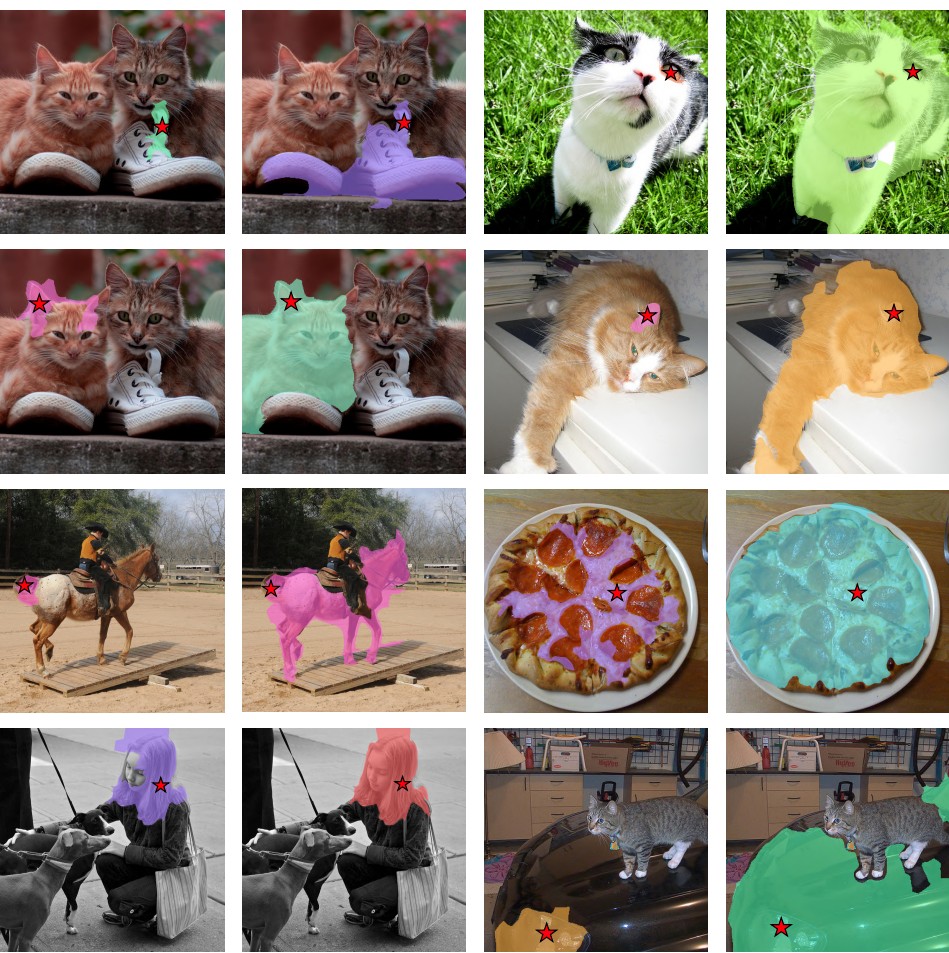

Figure 19: **Ambiguous instances:** If a cat's ear is marked, is the object the cat or the cat's ear? Images on the left: Grouping without explicit reference to "objectness". Images on the right: Grouping obtained using the "apostrophe-s" test.

the global mask is used as the consistency metric. Finally, the images that contain instances with a consistency value above a given threshold are selected and randomly sampled to create the evaluation set.

### E.3.1 WHAT CONSTITUTES AN OBJECT?

Determining the granularity of what qualifies as an object in a grouping task is often ambiguous. For instance, if a person's nose is highlighted, should the object be considered the nose alone, or the entire person? Both interpretations are valid, leading to potential inconsistencies.

Table 13: **Ablation study on depth prediction.** GPT-4o performs the best when curves are used as the visual marker.

| Method | Higher is better ↑ | | | | | Lower is better ↓ |
|---|---|---|---|---|---|---|
| | $\delta_1$ | $\delta_2$ | $\delta_3$ | $\rho$ | Accuracy | AbsRel |
| Curve | 0.550 | 0.822 | 0.935 | 53.75 | 70.43 | 0.332 |
| Rectangle | 0.534 | 0.807 | 0.931 | 51.68 | 69.28 | 0.341 |
| Point | 0.525 | 0.802 | 0.928 | 51.89 | 62.07 | 0.366 |

Table 14: **Oracle depth results** with different numbers of superpixels and comparisons made during chaining.

| Superpixels | Samples | Higher is better ↑ | | | | Lower is better ↓ |
|---|---|---|---|---|---|---|
| | | $\delta_1$ | $\delta_2$ | $\delta_3$ | $\rho$ | AbsRel |
| 100 | 200 | 0.571 | 0.774 | 0.863 | 0.83 | 0.528 |
| 100 | 400 | 0.597 | 0.785 | 0.867 | 0.86 | 0.514 |
| 200 | 200 | 0.571 | 0.773 | 0.867 | 0.83 | 0.501 |
| 200 | 400 | 0.593 | 0.788 | 0.869 | 0.86 | 0.502 |

To address this, we propose a prompting method that refines the granularity of "objectness." By instructing the model to interpret the highlighted instance as a possessive noun—expressed through the "apostrophe-s" structure—the model is encouraged to group coarser objects. For example, when prompted with "person's nose," the model is guided to interpret the object as the person, rather than the nose alone. This approach is illustrated in Fig. 19.

While this method is not universally effective, it often resolves ambiguity by clarifying the relationship between parts and wholes. We provide the full prompt in the supplementary material.

### E.4 DEPTH PREDICTION

We conduct an ablation study on the choice of visual markers in Tab. 13. Please also see Tab. 14 for additional oracle evaluations.

### E.5 SURFACE NORMAL PREDICTION

We conduct an ablation study on the choice of visual markers in Tab. 15.

### E.6 EXPERIMENTS WITH LLAMA

Unlike the other models, Llama employs different prompt chains for object detection, grouping and segmentation due to its current limitations with handling multiple images (Huggingface). Specifically:

- For object detection, we provide the full image with the corresponding grid cell marked instead of providing a crop of the grid cell with the full image.
- For semantic segmentation, we provide the full image with the corresponding superpixel marked, instead of providing a set of crops per superpixel.

Table 15: **Ablation study on surface normal prediction.** GPT-4o is relatively robust to different visual marker choices.

| Method | $\rho_x$ | $\rho_y$ | $\rho_z$ |
|---|---|---|---|
| Curve | -4.89 | 58.00 | 39.28 |
| Rectangle | -13.99 | 58.84 | 39.65 |
| Point | 2.42 | 51.26 | 39.59 |

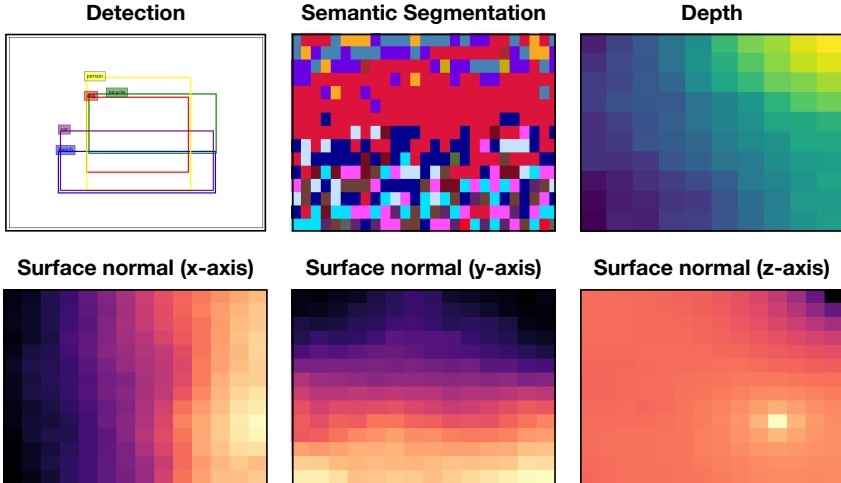

Figure 20: The blind guesses made by GPT-4o on different tasks.

- For grouping, we highlight the superpixel corresponding to the initial cluster in red and the superpixel of the query point in blue. The model is tasked with determining whether the blue region belongs to the same entity as the red region.

As with the other models, we experimented with multiple prompts and selected the best-performing one based on a smaller validation set across all tasks.

Surprisingly, Llama does well in segmentation despite not being provided with any crops. A comparative evaluation against other MFMs on a smaller subset using identical prompts is shown in Tab. 16. Notably, Llama surpasses all other models in this setup.

An additional interesting finding is Llama's unique capability to achieve a positive correlation in the $x$-direction for the surface normals task, a result not observed with the other non-reasoning MFMs.

Table 16: The performance of Llama compared with the other models on a smaller subset, in the absence of crops.

|      | Model | mIoU | Pixel Accuracy |
|------|-------|------|----------------|
|      | GPT-4o | 19.77 | 54.31 |
| MFMs | Gemini 1.5 Pro | 22.98 | 61.04 |
|      | Claude 3.5 Sonnet | 20.00 | 55.69 |
|      | Llama-3.2-90B | 25.86 | 61.66 |

### E.7 BLIND GUESS

As mentioned in Section 4, a useful way to analyze the potential biases of the MFM, and to gauge the degree to which it uses the visual content is a blind guess, or prompting the image with a blank image. In particular:

- For **object detection**, we ask the model to imagine the classes present. After this, we ask it to provide reasonable coordinates for the objects based on its world knowledge.
- For **semantic segmentation**, we mark a rectangle in a white image and force the model to predict a class. We ask the model to use the location to make an educated guess.
- For **depth**, we ask the model to imagine an indoor setting. We mark two rectangles and force the model to predict that one is at a greater depth than the other.
- For **normals**, we repeat the procedure for depth for each direction.

The results for GPT-4o are visualized in Fig. 20, and reveal several interesting insights.

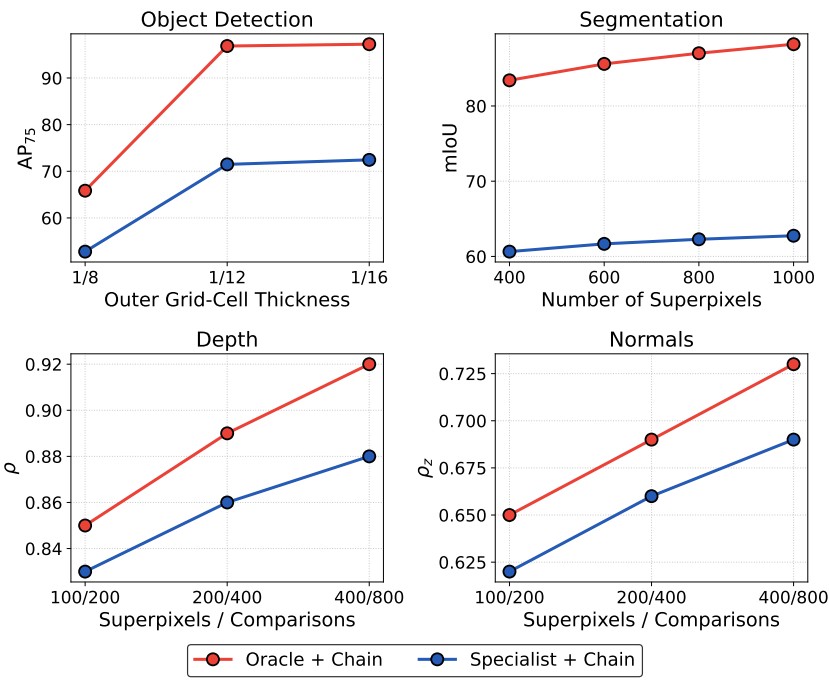

Figure 21: Performance improvements of the "Oracle + Chain" and "Specialist + Chain" baselines on full datasets when using a finer-grained prompt chain. Ten iterations were used for object detection.

- For **object detection**, the model chooses common classes like person and car. Additionally, it seems to grasp the relative sizes of objects reasonably well, as indicated by its tendency to make the car and the bench longer.

- For **semantic segmentation**, the model makes reasonable guesses. For instance, it guesses "sky-merged" and "airplane" at the top of the image, "person" near the middle, "dog," "cat," and "floor" near the bottom.

- For **depth prediction**, GPT-4o exhibits a "ceiling bias" and consistently infers that the top right corner is located at a greater relative depth. We observe that this bias is reflected in several of the model's predictions as well, where the ceiling is consistently assumed to be at a greater depth.

- For **surface normals**, the model uses the relative locations of the rectangles to form judgments. For instance, in the $x$ direction, it infers that the right rectangle aligns more towards the right. In the $y$ direction, it consistently infers that the bounding box at a greater $y$ coordinate aligns more with the positive $y$ direction. While Chain-of-Thought (CoT) reasoning is able to break this bias along the $y$ direction for GPT-4o, the left-right bias persists when actual images are presented.

## E.8 FINER-GRAINED PROMPT CHAIN

A natural question is whether the performance of our MFMs can be further enhanced by refining the granularity of the prompt chain. In other words, can we improve performance by increasing the number of superpixels and comparisons or by using thinner outer grid cells? To explore this, we first examine the effect of a finer-grained prompt chain on the "Oracle + Chain" and "Specialist + Chain" baselines. As shown in Fig. 21, these baselines exhibit steady performance improvements as the prompt chain is refined.

To determine whether this trend extends to the MFMs, we conducted a small-scale experiment using GPT-4o on the same tasks. As illustrated in Fig. 22, although GPT-4o shows modest improvements with a finer-grained prompt chain, its performance quickly saturates due to misclassifications. This

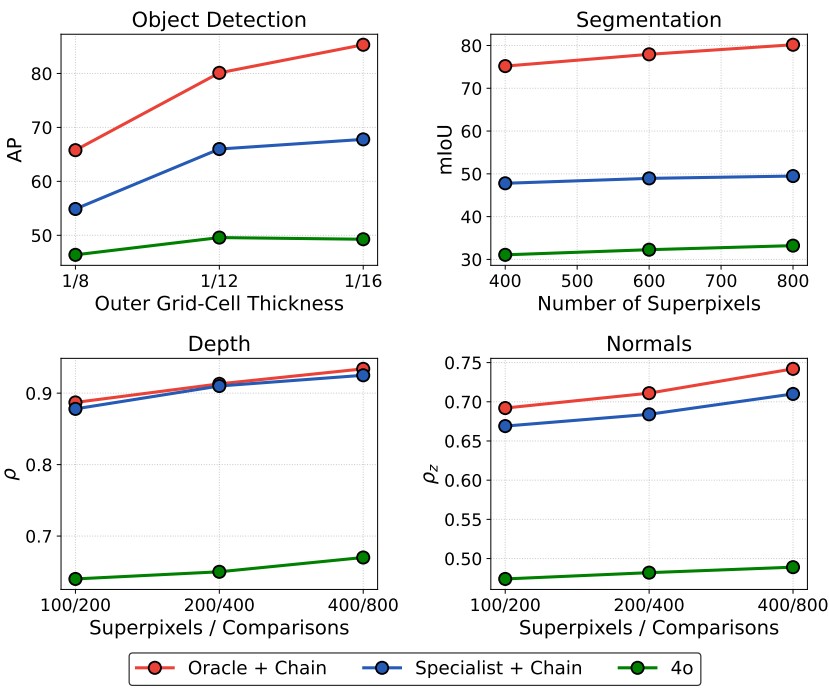

Figure 22: GPT-4o's performance improvements with a finer-grained prompt chain plateau, showing that our original settings are adequate.

observation supports our decision to adopt a coarser granularity for the MFMs, confirming that our original settings are sufficient to capture the performance gap.

We reiterate here that the control baselines in our paper serve as calibration tools rather than performance ceilings. They mitigate the sub-optimality of the prompting method and prevent exaggerated conclusions about MFMs significantly lagging behind specialists. Our benchmark is designed to be accurate in a relative and ordinal sense.

### E.9 PERFORMANCE CEILING FOR GEOMETRIC TASKS

To investigate whether the pipeline imposes a performance ceiling for the geometric tasks, we evaluated the depth and surface normal algorithms using increasingly fine-grained superpixelations. We varied the number of superpixels $N$ from 100 to 2000. To ensure a sufficient number of pairwise comparisons for the optimization, we set the number of pairwise comparisons at $N_{pairs} = 32 \times N$.

As shown in Tab. 17, increasing the granularity consistently improves the performance of the control baselines. Notably, the correlation metrics converge towards the "Dense" values (representing the performance obtained using raw pixel predictions without superpixel discretization). This confirms that the pipeline is capable of high-fidelity geometric recovery given sufficient granularity. We report correlation ($\rho$) as our algorithm reconstructs geometry purely from ordinal rankings.

We provide qualitative comparisons between the *Specialist + Chain* and *Oracle + Chain* reconstructions in Figures 23 and 24. The visuals are in agreement with the quantitative results: fine-grained geometric details are progressively recovered as the number of superpixels and comparisons increases, demonstrating that the prompt chaining mechanism itself does not inherently prevent high-fidelity estimation.

### E.10 IN-THE-WILD EVALUATIONS

Please see Fig. 27, 28 and 29 for qualitative evaluation of MFMs on in-the-wild samples (Flickr, 2024; Unsplash, 2024).

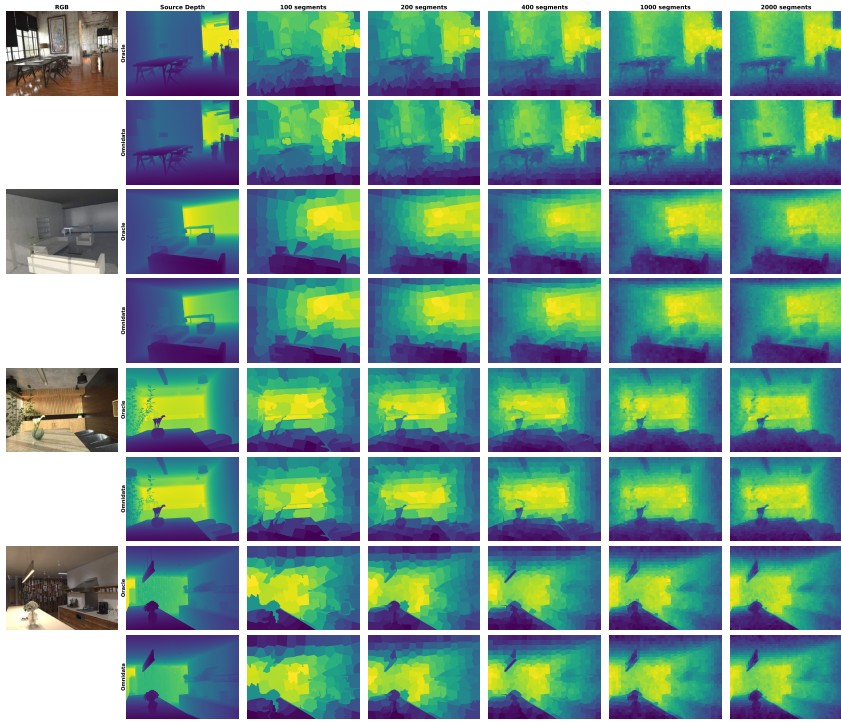

Figure 23: Increasing superpixel granularity in the depth prompt chain leads to the recovery of fine-grained features.

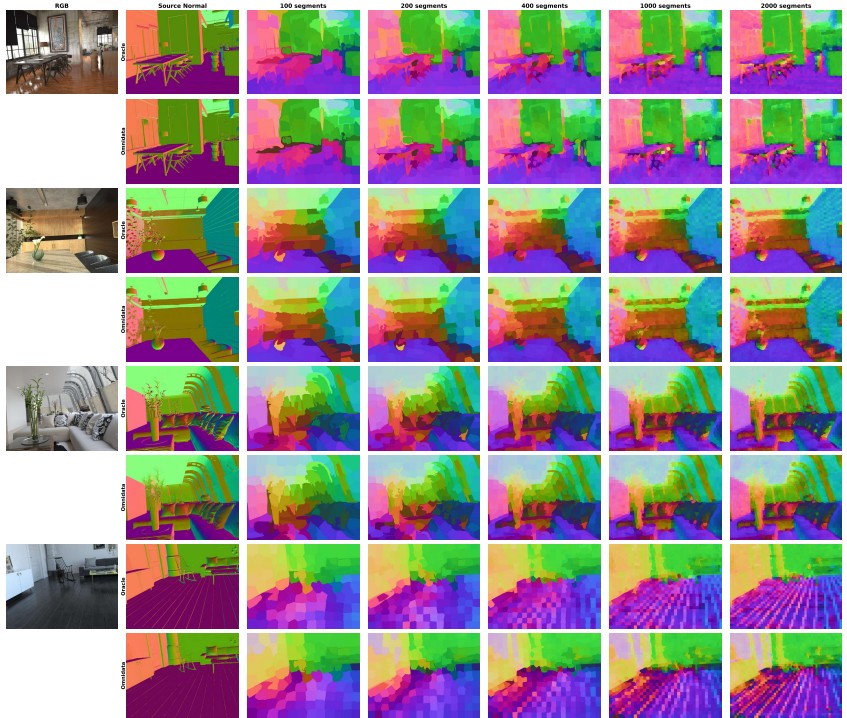

Figure 24: Similar to depth, increasing superpixel granularity in the surface normals prompt chain leads to a sharper normal map.

(a) Depth

| Metric: $\rho$ (↑) | Number of Superpixels | | | | | |
|---|---|---|---|---|---|---|
| Baseline | 100 | 200 | 400 | 1000 | 2000 | Dense |
| Oracle + Chain | 0.938 | 0.958 | 0.964 | 0.978 | 0.982 | 1.000 |
| Specialist + Chain | 0.924 | 0.943 | 0.952 | 0.958 | 0.962 | 0.969 |

(b) Surface Normals

| Metric: $\rho_z$ (↑) | Number of Superpixels | | | | | |
|---|---|---|---|---|---|---|
| Baseline | 100 | 200 | 400 | 1000 | 2000 | Dense |
| Oracle + Chain | 0.743 | 0.774 | 0.802 | 0.831 | 0.848 | 1.0 |
| Specialist + Chain | 0.713 | 0.735 | 0.751 | 0.762 | 0.769 | 0.805 |

Table 17: Effect of superpixel granularity on the Depth and Surface Normal Prediction baselines. We observe that the correlations approach the dense (raw pixel) performance ceiling.

# F   REASONING RESULTS

## F.1   ABLATING THE BATCH SIZE

As noted in Sec. 4.1, o4-mini is especially sensitive to the batch size used during inference. While our main evaluations used a batch size of 100 for consistency across models, we ablate this choice on a subset of 500 ImageNet samples in Fig. 25.

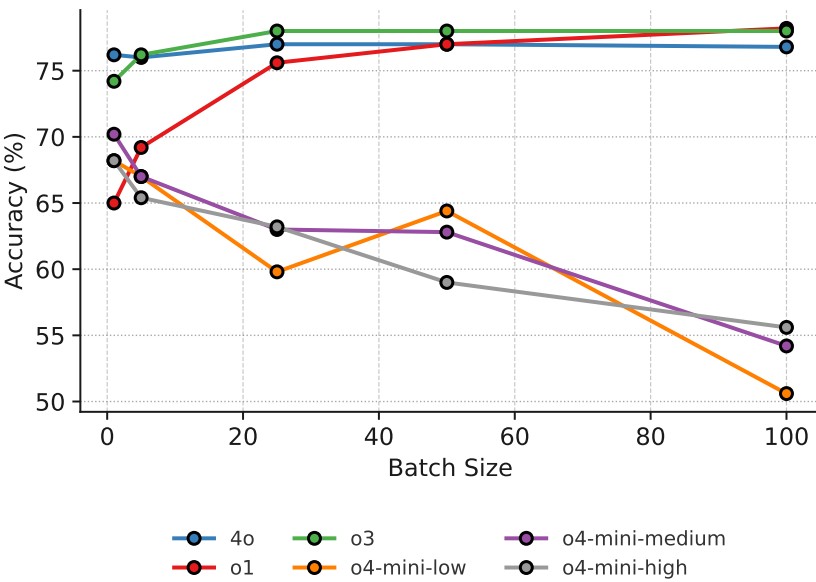

Figure 25: **Effect of batch size on classification accuracy.** We ablate the classification performance of o1, o3, o4-mini and GPT-4o on a subset of 500 ImageNet images under varying batch sizes. While o1 and o3 benefit from larger batch sizes, o4-mini shows a pronounced drop in accuracy.

As shown, o4-mini suffers a substantial degradation in classification accuracy as the batch size increases, across all reasoning effort settings. In contrast and surprisingly, both o1 and o3 demonstrate improved performance with larger batch sizes, consistent with trends observed in prior work on in-context learning for LLMs (Chen et al., 2023; Jiang et al., 2024b).

## F.2 Detailed Experiments

As noted in the main paper, we evaluate o1 and o3 on smaller, representative subsets of data. To construct informative subsets for classification, object detection, segmentation, and grouping, we compute the Kendall $\tau$ rank correlation between the performance rankings of non-reasoning models on candidate subsets and their full-dataset rankings, over multiple bootstrap runs. For each task, we choose the smallest subset size that meets a task-specific correlation threshold, yielding 1,000 samples for classification, 200 for detection, 50 for segmentation, and 30 for grouping. These subsets are deemed informative because they preserve the relative ranking of non-reasoning models.

For depth and surface normal prediction, we select the 10 most challenging samples for GPT-4o (those where it achieves the lowest Spearman correlation) due to higher evaluation costs.

Table 18: **Classification.** Accuracy scores on the standard classification datasets. o3 consistently outperforms GPT-4o, and achieves the highest overall scores. In contrast, o4-mini lags behind.

| Model | ImageNet | ImageNet-V2 | 2DCC | 3DCC | ImageNet-R | ImageNet-Sketch |
|---|---|---|---|---|---|---|
| o4-mini (low) | 50.7 | – | – | – | – | – |
| o4-mini (medium) | 58.4 | 49.2 | 35.6 | 36.4 | 58.2 | 46.4 |
| o4-mini (high) | 59.6 | – | – | – | – | – |
| o1 | 77.90 | 70.50 | 62.40 | 60.20 | 85.3 | 65.50 |
| o3 | 78.00 | 74.10 | 62.60 | 61.70 | 87.2 | 69.10 |
| GPT-4o | 76.70 | 72.10 | 63.80 | 61.40 | 86.2 | 66.10 |

Table 19: **Object Detection.** All models other than o4-mini achieve comparable performance, with o1 and o3 slightly outperforming GPT-4o.

| Model | $AP_{50}$ ($\uparrow$) | $AP_{75}$ ($\uparrow$) | AP ($\uparrow$) |
|---|---|---|---|
| o4-mini (low) | 48.40 | 29.71 | 27.65 |
| o4-mini (medium) | 47.56 | 29.77 | 27.11 |
| o4-mini (high) | 48.10 | 26.49 | 26.07 |
| o1 | 66.70 | 40.52 | 37.77 |
| o3 | 64.89 | 40.73 | 38.57 |
| GPT-4o | 64.11 | 42.61 | 38.17 |

Table 20: **Segmentation.** o1 and o3 outperform GPT-4o in both mIoU and pixel accuracy. o4-mini underperforms across all reasoning levels.

| Model | mIoU ($\uparrow$) | Pixel acc. ($\uparrow$) |
|---|---|---|
| o4-mini (low) | 29.67 | 63.94 |
| o4-mini (medium) | 28.86 | 63.82 |
| o4-mini (high) | 28.66 | 62.81 |
| o1 | 40.24 | 72.92 |
| o3 | 38.18 | 72.75 |
| GPT-4o | 35.66 | 69.59 |

Table 21: **Depth Prediction.** All models struggle with this difficult subset. However, o3 consistently outperforms the others across most metrics.

| Model | $\delta_1$($\uparrow$) | $\delta_2$($\uparrow$) | $\delta_3$($\uparrow$) | $\rho$($\uparrow$) | Accuracy($\uparrow$) | AbsRel($\downarrow$) |
|---|---|---|---|---|---|---|
| o4-mini (low) | 0.467 | 0.726 | 0.857 | -0.009 | 55.15 | 1.07 |
| o4-mini (medium) | 0.465 | 0.711 | 0.852 | 0.070 | 56.85 | 0.953 |
| o4-mini (high) | 0.462 | 0.724 | 0.864 | -0.029 | 53.65 | 1.090 |
| o1 | 0.456 | 0.716 | 0.873 | 0.079 | 57.20 | 0.914 |
| o3 | 0.490 | 0.738 | 0.855 | 0.250 | 63.50 | 0.819 |
| GPT-4o | 0.460 | 0.732 | 0.881 | -0.050 | 52.00 | 1.121 |

The performance of o1, o3, GPT-4o, and o4-mini, under varying levels of reasoning effort, is summarized in Tables 18 through 23, covering classification, object detection, segmentation, grouping, depth, and surface normal prediction.

On semantic tasks, all models perform comparably, with o1 and o3 showing slightly stronger results in classification, object detection, and segmentation. For geometric tasks, performance drops across the board due to the difficulty of the selected samples. However, the reasoning models consistently outperform GPT-4o. In particular, all reasoning models achieve positive correlation along the horizontal axis in surface normal prediction, correcting a common failure mode in GPT-4o (see Fig. 8, where the horizontal gradient is flipped). Qualitative comparisons for all tasks are shown in Fig. 8.

Table 22: **Grouping.** GPT-4o outperforms all reasoning-based models on this semantic task. Among the reasoning models, o3 performs best.

| Model | mIoU($\uparrow$) |
|---|---|
| o4-mini (low) | 47.41 |
| o4-mini (medium) | 44.59 |
| o4-mini (high) | 46.36 |
| o1 | 52.61 |
| o3 | 55.06 |
| GPT-4o | 59.64 |

Table 23: **Surface Normals.** Reasoning models outperform GPT-4o, particularly along the horizontal ($x$) direction, where GPT-4o shows a strong negative correlation. o3 achieves the highest scores across all axes.

| Model | $\rho_x$ | $\rho_y$ | $\rho_z$ |
|---|---|---|---|
| o4-mini (low) | 0.18 | 0.12 | 0.30 |
| o4-mini (medium) | 0.39 | 0.24 | 0.32 |
| o4-mini (high) | 0.38 | 0.23 | 0.31 |
| o1 | 0.24 | 0.23 | 0.19 |
| o3 | 0.48 | 0.36 | 0.28 |
| GPT-4o | -0.30 | 0.09 | 0.05 |

# G    ADDITIONAL EXPERIMENTS WITH 4O IMAGE GENERATION

## G.1    PROMPTING METHODOLOGY

As discussed in the main paper, GPT-4o generates full image recreations rather than edits, often resulting in spatial misalignments. To enable consistent comparisons, we first zero-pad all input images to square dimensions and, after generation, crop the relevant regions to align with the original input. Below, we outline task-specific prompting details. Preliminary quantitative results for grouping, depth, and surface normals, evaluated at full scale, are provided in App. G.2.

**Grouping.** The prompt used for object-based segmentation is shown in Listing 1. The model is given an image with a red point marking a location on an object and is instructed to return the same image with the entire object filled in solid red. To extract the predicted mask, we use HSV thresholding and post-process it by retaining only the largest connected component, which effectively removes small artifacts and hallucinated regions. Although this simple postprocessing often yields reasonable masks, more advanced postprocessing techniques could improve results further. We leave such refinements to future work.

```
You are given an input RGB image where a small red circle marks a point
    on an object.
Your task is to return the **exact same image**, but with the **entire
    object that contains the marked point filled in solid red**.
Do not add any other markings, text, or overlays; only apply the red fill
     to the object.
```

Listing 1: Grouping prompt for GPT-4o image generation

**Depth.** The prompt used for depth prediction is provided in Listing 2. The model is instructed to produce a grayscale rendering of the input image, where closer regions appear darker and farther regions lighter.

```
Generate a **pure grayscale depth map** from the input image. The
    grayscale values must encode depth as follows:

- **White (255 intensity)**  represents points that are **closest** to
    the camera (minimum depth).
- **Black (0 intensity)** represents points that are **farthest** from
    the camera (maximum depth).
- All other points must be shaded **monotonically between black and white
    **, based solely on their distance from the camera.

This map must not contain any colors, textures, or artistic effects: only
     smooth grayscale transitions that accurately reflect increasing
    depth, with darker shades at greater distances.
```

Listing 2: Depth prediction prompt for GPT-4o image generation

**Surface Normals.** The prompt used for surface normal prediction is shown in Listing 3. The model is asked to generate a surface normal map in the standard RGB encoding used in computer graphics. Conventionally, red corresponds to the left–right ($x$) axis (left = 0, right = 1), green to the up–down ($y$) axis (down = 0, up = 1), and blue to the depth ($z$) axis (inward = 0, outward = 1).

```
You are a vision model that, given an input RGB image, must predict a per
    -pixel surface normal map and render it as an RGB image using the
    standard normal-map color scheme.

Requirements:
1. **Output format**
- Directly generate an image.
- Produce a raw image (same dimensions as input) whose pixel colors
    encode the normals as above.
- Do **not** add any annotations, text overlays, or alpha channels: only
    the RGB channels.
2. **Normal-map encoding**
- For each pixel estimate its surface normal vector
- Display the orientation of the surface normal vector using the standard
     color scheme used in computer graphics.
```

Listing 3: Normals prediction prompt for GPT-4o image generation

### G.1.1 PROMPT SENSITIVITY

We observe that GPT-4o's image generation is highly sensitive to changes in the prompt. To illustrate this brittleness, we test two modified prompts. For depth prediction, we invert the color mapping in Listing 2, asking the model to render *near* points as dark and *far* points as white. For surface normals, rather than producing a single RGB normal map, we prompt the model for three separate grayscale images, each representing alignment with one of the x, y, or z axes, similar to our prompt-chaining setup. As shown in Tab. 24, these small changes lead to a substantial degradation in performance.

Table 24: **Performance for different prompts.** Comparison between the original and an altered prompt for both depth prediction and surface normal prediction.

(a) Depth prediction

| Prompt | $\delta_1$ ($\uparrow$) | $\delta_2$ ($\uparrow$) | $\delta_3$ ($\uparrow$) | $\rho$ ($\uparrow$) | AbsRel ($\downarrow$) |
|---|---|---|---|---|---|
| Original | 0.562 | 0.849 | 0.942 | 0.448 | 0.303 |
| Altered | 0.399 | 0.694 | 0.854 | -0.12 | 0.474 |

(b) Surface normal prediction

| Prompt | $\rho_x$ | $\rho_y$ | $\rho_z$ |
|---|---|---|---|
| Original | 0.31 | 0.35 | 0.14 |
| Altered | -0.15 | 0.38 | -0.23 |

### G.2 PRELIMINARY QUANTITATIVE ANALYSIS

The full-scale quantitative results are shown in Tab. 25. While the performance is non-trivial, it currently falls short of what is achieved by GPT-4o using the prompt chain. We view the refinement of prompts and decoding strategies for image generation as an important direction for future work.

Notably, grouping predictions are impacted by spatial misalignment, hallucinated regions, and incorrect markings. Depth predictions, on the other hand, occasionally suffer from an inverted rendering of the depth gradient, which significantly affects correlation-based metrics. We showcase representative qualitative results for grouping, depth, and surface normals in Fig. 9, highlighting both successes and common failure modes.

Table 25: GPT-4o image generation performance across three tasks.

| Task | Metric | Value ($\uparrow$) |
|---|---|---|
| Grouping | mIoU | 28.14 |
| Depth Prediction | $\delta_1$ | 0.485 |
| | $\delta_2$ | 0.735 |
| | $\delta_3$ | 0.848 |
| | $\rho$ | 0.52 |
| | AbsRel ($\downarrow$) | 0.575 |
| Surface Normal Prediction | $\rho_x$ | 0.09 |
| | $\rho_y$ | 0.44 |
| | $\rho_z$ | 0.17 |

## H PROMPT SENSITIVITY ANALYSIS

In Fig. 26, we evaluate the non-reasoning models for each task considering different prompting techniques. We observe that GPT-4o generally shows lower sensitivity to different prompts on most of the tasks compared to other MFMs. For surface normals, we interestingly observe that the predictions greatly improve in the $y$ and $z$ directions, when GPT-4o and Claude are asked to reason in the prompt (see Tab. 26).

## I PROMPTING COSTS

At the time of writing, the API pricing for GPT-4o (`gpt-4o-2024-08-06`) was $2.50 per million input tokens and $10.00 per million output tokens. For Gemini 1.5 Pro (`gemini-1.5-pro-001`), the corresponding rates were $3.50 (input) and $10.50 (output), and for Claude 3.5 Sonnet (`claude-3-5-sonnet-20240620`), $3.00 and $15.00, respectively. In contrast, lower-cost models like Gemini 2.0 Flash (`gemini-2.0-flash-001`) and o4-mini (`o4-mini-2025-04-16`)

Table 26: **Prompt sensitivity for surface normal prediction.** Correlations under five different prompts for GPT-4o and Claude 3.5 Sonnet. CoT prompting (**in bold**) greatly improves $\rho_y$ and $\rho_z$.

| Model | Prompt | $\rho_x$ ($\uparrow$) | $\rho_y$ ($\uparrow$) | $\rho_z$ ($\uparrow$) |
|---|---|---|---|---|
| GPT-4o | Prompt 1 | -0.36 | -0.63 | 0.04 |
| | Prompt 2 | -0.29 | -0.45 | -0.10 |
| | **Prompt 3** | **0.07** | **0.55** | **0.44** |
| | **Prompt 4** | **-0.08** | **0.65** | **0.43** |
| | **Prompt 5** | **-0.34** | **0.45** | **0.37** |
| Claude 3.5 Sonnet | Prompt 1 | -0.34 | -0.56 | 0.00 |
| | Prompt 2 | -0.18 | -0.08 | -0.11 |
| | **Prompt 3** | **-0.09** | **0.68** | **0.41** |
| | **Prompt 4** | **-0.06** | **0.66** | **0.35** |
| | **Prompt 5** | **-0.06** | **0.56** | **0.35** |

were priced at \$0.10/\$0.40 and \$1.10/\$4.40, respectively. The reasoning models `o1-2024-12-17` and `o3-2025-04-16` were substantially more expensive at \$15.00/\$60.00 and \$10.00/\$40.00, respectively.

The costs for the scaled-up experiments are documented in Tab. 27, and the prompting costs for the reasoning model are presented in Tab. 28. The primary reason for cost fluctuations across tasks is the way each MFM tokenizes images. The notably higher costs for object detection with Gemini 1.5 Pro stem from the independent calls required in the prompt chain. For reasoning models, especially on the surface normal task, a major contributor is the large number of generated reasoning tokens. The availability of highly affordable models like Gemini 2.0 Flash, for which **our entire evaluation cost approximately \$50**, demonstrates that such benchmarking is becoming increasingly accessible.

These costs reflect the constraints of current APIs and are not indicative of how such tasks would be solved in practical deployments. As discussed in the main paper, our framework is intended for a standardized one-time evaluation (and not for efficient task execution) with MFMs.

Table 27: Prompting costs for scaled-up experiments (in \$).

| Task | GPT-4o | Gemini 1.5 Pro | Claude 3.5 Sonnet | Gemini 2.0 Flash |
|---|---|---|---|---|
| Classification | 223.8 | 298.6 | 142.8 | 9.7 |
| Object Detection | 185.8 | 610.8 | 155.0 | 18.1 |
| Semantic Segmentation | 232.1 | 450.1 | 227.9 | 14.0 |
| Grouping | 22.1 | 47.4 | 42.0 | 1.0 |
| Depth | 57.4 | 52.4 | 198.2 | 3.6 |
| Normals | 130.1 | 50.1 | 209.9 | 3.9 |

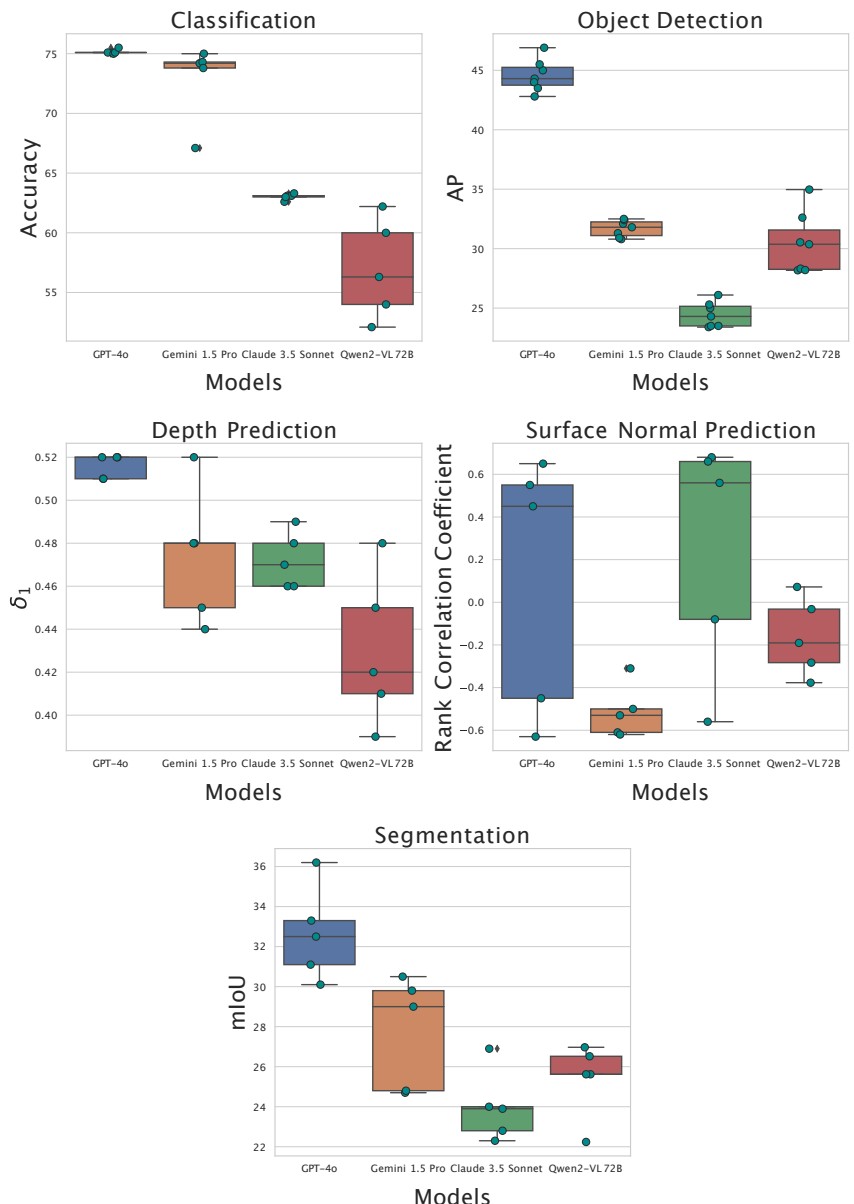

Figure 26: Sensitivity of MFMs to different prompting techniques. We observe that GPT-4o showcases a lower sensitivity on most tasks compared to other MFMs.

Table 28: Prompting costs for the reasoning models (in \$).*Reported costs are for experiments conducted on the subset.

| Task | o1* | o3* | o4-mini |
|---|---|---|---|
| Classification | 41.0 | 22.5 | 50.0 |
| Object Detection | 220.0 | 104.0 | 102.2 |
| Semantic Segmentation | 200.0 | 96.0 | 115.0 |
| Grouping | 82.2 | 48.9 | 25.0 |
| Depth | 96.4 | 31.5 | 62.0 |
| Normals | 306.2 | 85.4 | 194.0 |

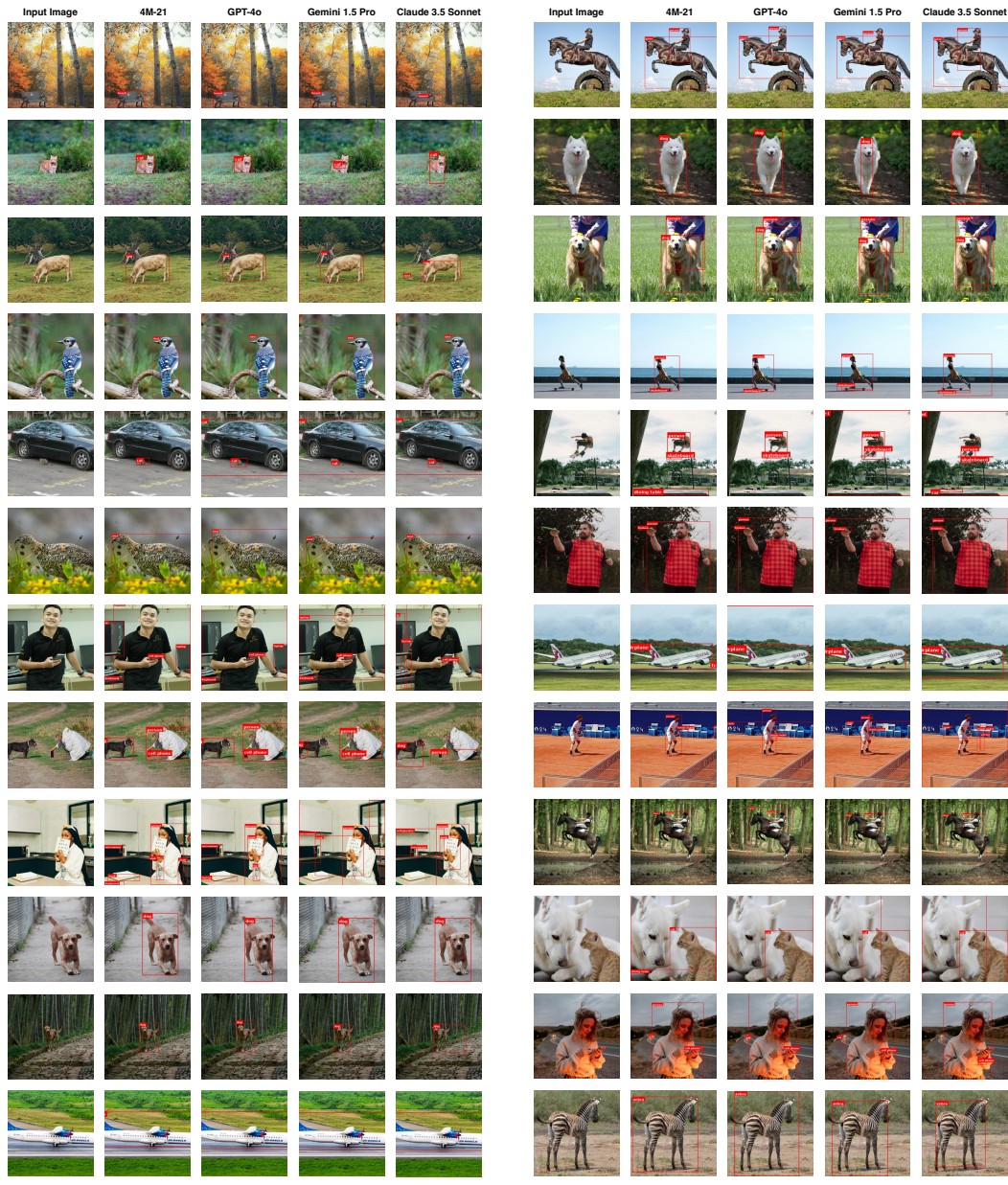

Figure 27: Qualitative results of evaluating MFMs for object detection on in-the-wild examples (Flickr, 2024; Unsplash, 2024). We compare against 4M-21 (Bachmann et al., 2024) as a vision specialist.

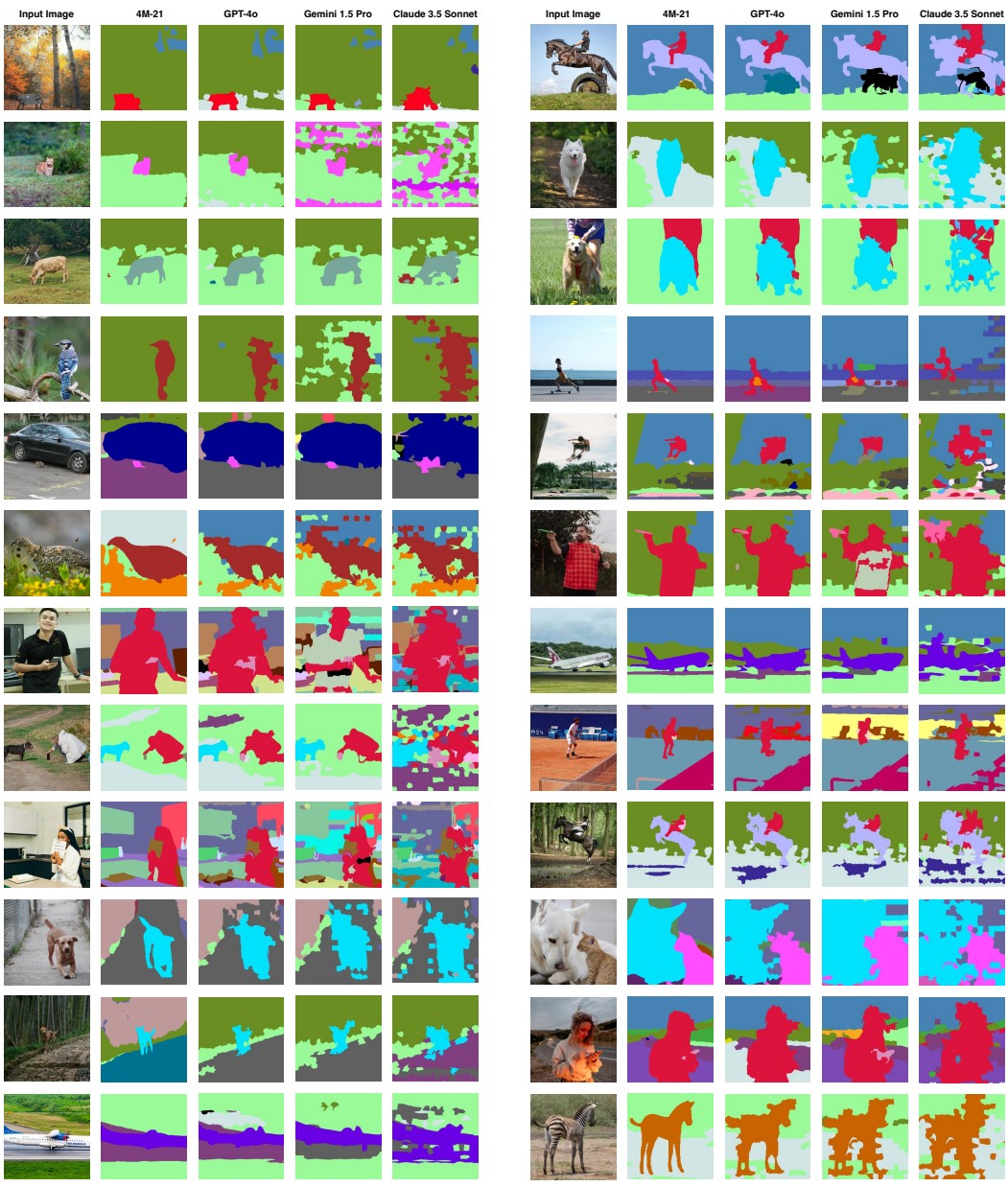

Figure 28: Qualitative results of evaluating MFMs for semantic segmentation on in-the-wild examples (Flickr, 2024; Unsplash, 2024). We compare against 4M-21 (Bachmann et al., 2024) as a vision specialist.

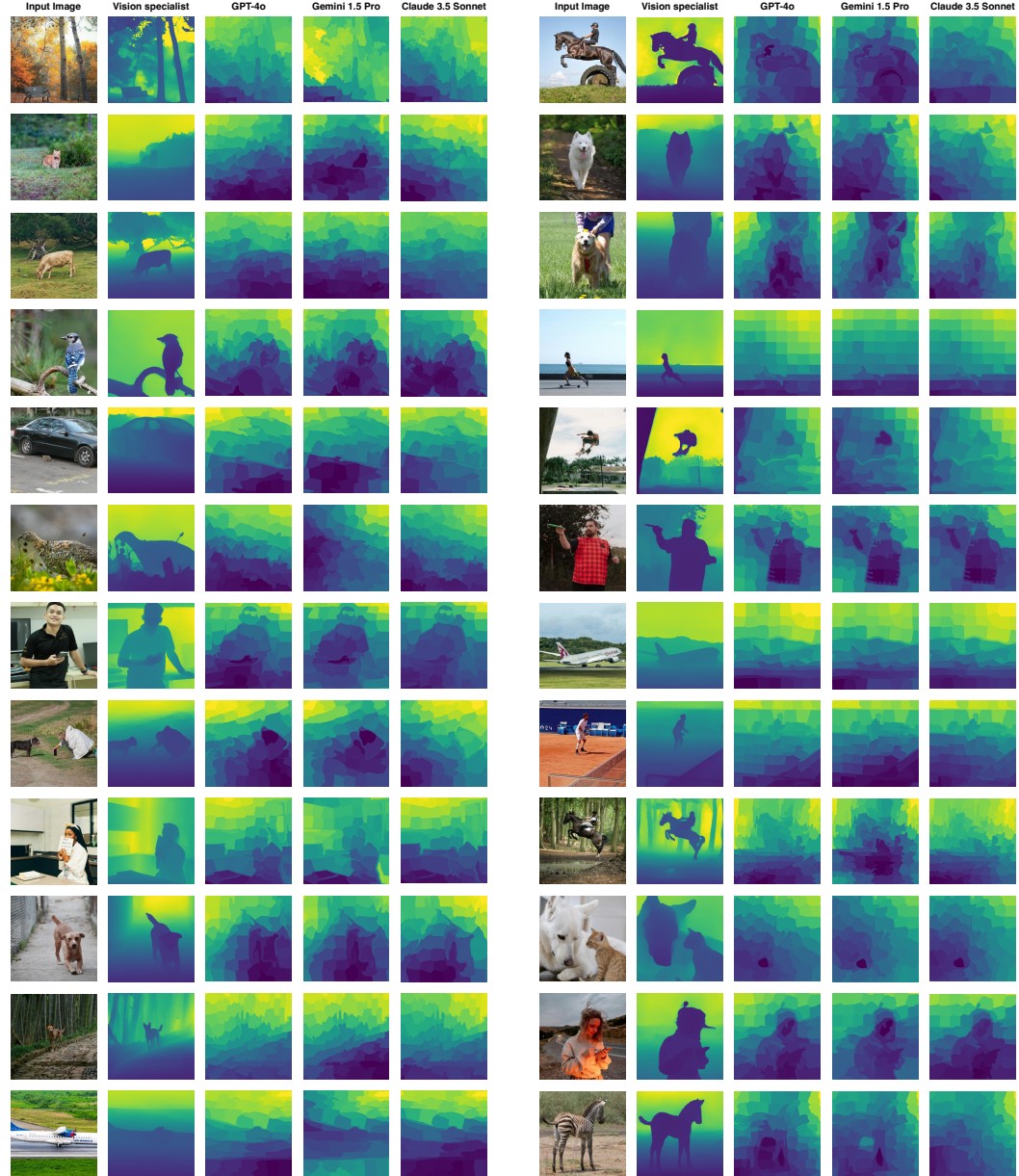

Figure 29: Qualitative results of evaluating MFMs for depth prediction on in-the-wild examples (Flickr, 2024; Unsplash, 2024). We compare against the Omnidata (Eftekhar et al., 2021) depth estimator as a vision specialist.

