# OpenReview forum: "How Well Does GPT-4o Understand Vision? Evaluating Multimodal Foundation Models on Standard Computer Vision Tasks"
_ICLR.cc/2026/Conference — ICLR 2026 Poster_

### Official Review · Reviewer_GMJw · 2025-10-31

**Soundness:** 4
**Presentation:** 3
**Contribution:** 3
**Rating:** 8
**Confidence:** 4

**Summary:**

This paper benchmarks VLMs on standard computer vision tasks (semantic segmentation, object detection, image classification, depth and surface normal prediction) using established datasets. It developed a novel method to translate vision tasks into text-promptable, API-compatible formats via prompt chaining. It draws a few interesting findings: VLMs are not close to the state-of-the-art specialist
models at any tasks; GPT-4o performs the best among non-reasoning models, securing the top position in 4 out of 6 tasks.

**Strengths:**

1. The authors develop a very clever way to prompt the VLMs to perform dense prediction tasks by using superpixels. They also find a better way to prompt object detection through iterative cropping and splitting the image into grids. This new technique creates the new possibility of systematically benchmarking the spatial and semantical understanding capability of VLMs, and the results are interesting.
2. The paper is well-written. The experiments are extensive, covering 6 fundamental vision tasks and the state-of-the-art VLMs, both closed and open-sourced ones.

**Weaknesses:**

1. Some of the specialist models are outdated. This could make the readers think that the gaps between VLMs and SOTA models are smaller than they actually are. Examples: the authors should compare with SAM2 in semantic segmentation (Tab 4); compare with Lotus [1] and Moge2 [2] for depth and normal map estimation.
2. For depth and normal map estimation, it is not clear how many random pairs are needed for a good convergence of the optimization algorithm. The results of depth and normal in Fig 3 also look very coarse, probably due to the granularity of the superpixels. It would be interesting to see if higher quality depth/normal map will emerge if we use more superpixels and/or more pairs.

[1] Lotus: Diffusion-based Visual Foundation Model for High-quality Dense Prediction, ICLR 2025

[2] MoGe-2: Accurate Monocular Geometry with Metric Scale and Sharp Details, Arxiv 2025

**Questions:**

N/A

---

> ### Author Response · Authors · 2025-11-26
> **Response to reviewer GMJw**
>
> We thank reviewer GMJw for their time and constructive feedback. We address the main concerns below.
>
> > Examples: the authors should compare with SAM2 in semantic segmentation (Tab 4); compare with Lotus and Moge2 for depth and normal map estimation.
>
> We agree, as SOTA specialist models are a fast moving goal. To address this, we have added experiments with other specialists. We now include SAM2 for segmentation, and Lotus, MoGe2, and DSINE for depth and surface normals under both their native evaluation protocols and our chaining setup. As expected, the chained variants of these models substantially outperform the MFMs under the same constraints. We will add these additional results and any obvious established new model to the camera-ready version.
>
> - Depth
>
> | Model              | $δ₁$ |$δ₂$ | $δ₃$ | $ρ$ |
> |--------------------|------------|------------|------------|--------|
> | Lotus              | 0.776      | 0.861      | 0.913      | 0.93   |
> | Lotus + Chain  | 0.578  | 0.779  | 0.866  | 0.82 |
> | Moge               | 0.795      | 0.859      | 0.903      | 0.90   |
> | Moge + Chain   | 0.576  | 0.774  | 0.861  | 0.82 |
> | Strongest MFM  | 0.467  | 0.718  | 0.841  | 0.58 |
>
> - Surface normals
>
> | Model             |$ρ_x$ | $ρ_y$ | $ρ_z$ |
> |-------------------|----------|----------|----------|
> | Lotus             | 0.77     | 0.80     | 0.75     |
> | Lotus + Chain | 0.64 | 0.70 | 0.54 |
> | Moge              | 0.80     | 0.80     | 0.77     |
> | Moge + Chain  | 0.64 | 0.69 | 0.56 |
> | DSINE             | 0.76     | 0.79     | 0.73     |
> | DSINE + Chain | 0.63 | 0.69 | 0.55 |
> | Strongest MFM | 0.22 | 0.61 | 0.46 |
>
> - Grouping
>
> | Model             | mIoU  |
> |-------------------|-------------------|
> | Sam-2             | 80.87             |
> | Sam-2 + Chain | 72.35         |
> | Strongest MFM | 59.06         |
>
>
> > Convergence of optimization / Granularity
>
> We address convergence and granularity in **App. E.8 (Fig. 22)**. We vary the number of superpixels from 100 to 400 on a subset of the data and observe that performance **saturates quickly**: increasing the number of superpixels yields only marginal improvements, indicating that beyond a certain granularity, the dominant source of error is the MFM’s predictions rather than optimization failure. The gap between specialists and MFMs remains large across granularities, confirming that our chosen superpixel resolution is not an unreasonable cost–performance trade-off as it preserves the relative ranking.

---

### Official Review · Reviewer_9qaY · 2025-10-31

**Soundness:** 3
**Presentation:** 3
**Contribution:** 3
**Rating:** 8
**Confidence:** 5

**Summary:**

This paper presents a comprehensive benchmark evaluating the visual understanding capabilities of state-of-the-art multimodal foundation models (MFMs), including GPT-4o, Gemini 1.5/2.0, Claude 3.5 Sonnet, Qwen2-VL, and Llama 3.2, on a suite of standard computer vision tasks: classification, object detection, semantic segmentation, grouping, depth estimation, and surface normal prediction. Recognizing the challenge that most MFMs are text-based and do not natively output dense or structured visual predictions, the authors introduce a systematic “prompt chaining” framework that decomposes vision tasks into API-compatible sub-tasks. The results demonstrate that while MFMs are decent generalists—especially on semantic tasks like classification and segmentation—they lag substantially behind state-of-the-art specialist vision models, and struggle most with geometric (3D) tasks. The paper includes both quantitative and qualitative analyses, explores prompt sensitivity, and openly acknowledges the limitations of the approach.

**Strengths:**

**1) Thorough and Timely Benchmarking:** The paper provides one of the most exhaustive and systematic evaluations to date of leading MFMs' visual understanding, moving well beyond the customary VQA or captioning settings to address a broader spectrum of fundamental computer vision tasks.

**2) Task Translation via Prompt Chaining:** The “prompt chaining” framework is thoughtfully designed, enabling apples-to-apples comparisons between text-based MFMs and vision specialists through structured sub-task decomposition. Figure 2 effectively visualizes the design for converting complex tasks like depth and segmentation into textual queries, making the framework transparent and reproducible.

**3) Comprehensive Model and Task Coverage:** The analysis involves a diverse set of MFMs (both closed- and open-weight) across six core vision tasks and multiple datasets, situating the current state of MFMs within established benchmarks (COCO, ImageNet, Hypersim, etc.), including corruption and robustness variants. Table 1 and Table 3–6 substantiate this comprehensiveness.

**4) Control Baselines and Calibration:** The empirical results are carefully calibrated with various baselines—(a) top specialist models, (b) specialists subjected to the same chaining and superpixel constraints, (c) oracle variants, and (d) blind guess. This tightens the attribution of observed deficits to either model limitations or task translation artifacts.

**Weaknesses:**

**1) Prompt Chaining Overhead and Realism:** The proposed evaluation relies on decomposing tasks into a large number of textual API calls, which is computationally expensive (as noted in App. I). While the paper claims this is for benchmarking only, the translation introduces additional sources of potential error (task granularity, superpixel boundaries, etc.), and the extent to which these reflect real “model” limitations is not exhaustively disentangled. For instance, in Figure 2 and related descriptions, there is acknowledgment that chaining is not optimal—yet the effect of granularity and design choices is mostly probed by coarse ablations, potentially underestimating the ceiling performance of some MFMs.

**2) Limited Exploration of Advanced Prompting or Visual Tools:** There is a missed opportunity to rigorously evaluate more advanced or recently proposed visual prompt engineering techniques (e.g., interactive alignment, visual rulers, or interactive markers for geometric tasks, as briefly mentioned in App. E.1, Figure 15) across all models as a systematic solution for object localization or dense prediction. The decision to use superpixel-based batch querying is pragmatic but may limit the apparent ability of models on fine-structured tasks.

**3) Insufficient Error Analysis on Geometric Tasks:** While the paper establishes that MFMs fare significantly worse on geometric tasks (Tables 5 and 6; Figure 3), there is minimal deep dive into why—for example, what kinds of 3D/normal ambiguities (left-right, scale, out-of-plane rotation) are most problematic, or whether cues are absent due to model design, pretraining data, or prompt interface. Figure 3 and Table 6 show low/negative correlations for some directions, but the causal factors or failure modes are not dissected in detail.

**Questions:**

Please see weaknesses.

---

> ### Author Response · Authors · 2025-11-25
> **Response to reviewer 9qaY**
>
> We thank reviewer 9qaY for their time and constructive feedback. We address the main concerns below.
>
> > large number of textual API calls, which is computationally expensive
>
> We agree that prompt chaining is **not** intended for deployment-time inference, but emphasize that **our framework is instead intended for one-time evaluation** (L470). In this context, higher query counts are an acceptable trade-off to obtain a rigorous assessment of visual capability. Additionally, with rapid improvements in efficiency, such as the ~$50 cost for a full Gemini 2.0 Flash run (Appendix I), the benchmark remains accessible for the community.
>
> > The translation introduces additional sources of potential error (and the extent to which these reflect real “model” limitations is not exhaustively disentangled.
> >
>
> We share the concern about disentangling “translation error” from genuine model limitations. Attempting to control for this is exactly why the **Oracle+Chain** and **Specialist+Chain** baselines are central to our design:
>
> - **Oracle+Chain** upper-bounds performance given perfect answers under our chaining procedure.
> - **Specialist+Chain** measures the performance drop induced by the chaining itself when applied to strong vision specialists.
>
> These baselines **quantify the cost of chaining** and show that MFMs still perform significantly below specialists even when both operate under the same constraints. Therefore, the remaining performance gap can be attributed to the MFMs’ visual understanding rather than the translation mechanism.
>
> > Limited Exploration of Advanced Prompting or Visual Tools
> >
>
> We agree that this is an exciting frontier. Within the scope of this paper, we performed a systematic but bounded exploration of visual prompting variants to pick fair and reasonable ones (App. E.1, E.2):
>
> - For object detection, we evaluated three types of visual rulers and grid prompts (Fig. 15, Table 9).
> - For segmentation, we explored various marker types (Fig. 16, Table 11).
> - We further tested alternative marker styles (Fig. 18)
>
> Across these settings, we did not find a single strategy that substantially changed the overall conclusions or closed the gap to specialists. We therefore chose the best-performing prompt variants from this search and treat them as a strong lower bound on current MFM capability. The exact numbers are obviously dependent on the choice of promoting method, and it is fair to expect the more advanced ones now or the in the future to improve the results of the MFMs. However, our observed trends are likely more stable than the exact numbers since they were chosen systemically with reasonable efforts and without a bias toward one method.
>
> > Insufficient Error Analysis on Geometric Tasks
> >
>
> We agree that understanding *why* MFMs fail on geometry is crucial. While the closed nature of these models prevents direct inspection of training data or internal representations, we analyze specific failure modes and ambiguities via:
>
> - **Blind Guess Analysis (App. E.7):** When forced to guess depth without any explicit geometry prompts, models exhibit a systematic **“ceiling bias”**, tending to interpret pixels higher in the image as farther away. We observe this pattern repeatedly across images.
> - **Coordinate Ambiguities (App. H):** As the reviewer notes, we analyze how several models invert the x-axis in surface normals (confusing left-facing and right-facing surfaces), which manifests as negative correlations in Table 6. We find similar ambiguities along the y-axis; importantly, our chain-of-thought prompting reduces the y-axis ambiguity (Table 25), demonstrating that there is not a complete absence of geometric cues.
>
> We will expand the discussion in the the camera-ready to better address this point.

---

### Official Review · Reviewer_hd9G · 2025-10-31

**Soundness:** 2
**Presentation:** 2
**Contribution:** 1
**Rating:** 2
**Confidence:** 5

**Summary:**

This paper benchmarks multimodal foundation models for standard CV tasks. They proposed an interesting way to avoid using API calls for promptable computer vision tasks. The whole evaluation tasks covers 2 categories, semantic and geometrical understanding, and there are 6 sub-tasks in total. GPT-4o achieves an overall best performance, but is still lagging far behind CV models. For geometric tasks, under the promtable format, all the MFM failed to work.

**Strengths:**

- The promptable task design itself is interesting and is effective for semantic understanding tasks.
- This paper is well written and easy to follow
- Experimental evaluations are comprehensive.

**Weaknesses:**

The biggest problem is, the pipeline of the geometric understanding tasks (depth and surface normal) does not work at all for MFMs:
- In Figure 3, neither depth nor surface normal works as expected, making it not surprising that all the models lagged far behind CV models.
- The root cause of the mismatch is the ranking algorithm. It is impossible to get the numeric results for depth and surface normal purely from ranks.
- Without these two geometric tasks, all 4 tasks are basically semantic understanding tasks, with the performance of all the compared models highly correlated.

Other minor weaknesses:
- The observations did not provide key findings, unless the ranking. Some interesting points may be: why the reasoning models work worse on semantic understanding?
- The metrics for surface normal and depth are not the most popular ones.
- Results with GPT-4o image generation is confusing and did not fit the whole story.

**Questions:**

- How will random superpixelizations/grid seeds affect the performance?
- Could other CV tasks also be evaluated in the same manner, such as pose estimation, edge detection?

---

> ### Author Response · Authors · 2025-11-25
> **Response to reviewer hd9G**
>
> We thank reviewer hd9G for their time and constructive feedback. We address the main concerns below.
>
> > “the pipeline of the geometric understanding tasks does not work at all… ”
>
> We do not agree that the geometric pipeline “does not work.” Regarding the concern that the ranking algorithm is the root cause of the mismatch, we note two points. First, this optimization framework is an established methodology for recovering structure from ordinal constraints [3]. Second, the entire evaluation is calibrated using our `Oracle+Chain` and `Specialist+Chain` baselines. **These baselines utilize the identical superpixelization, ranking, and optimization machinery as the MFMs yet achieve markedly higher performance.** This demonstrates that the pipeline is effective, and the **observed gap reflects the MFMs’ genuine geometric limitations rather than a failure of the algorithm. Though the pipeline has to work with the fact that the models don’t output geometry directly, the evaluation pipeline is carefully designed and controlled to account for this externally imposed reality as much as possible.**
>
> > It is impossible to get the numeric results for depth and surface normal purely from ranks
>
> Our evaluation is explicitly based on **relative** geometry, not direct metric reconstruction. For depth, we obtain ordinal judgments (e.g., “which superpixel is closer?”) to optimize a global depth map. To evaluate this relative prediction, we apply a least-squares alignment (scale and shift) to the ground truth, a standard practice in monocular depth literature [1, 2], before computing metrics.
>
> The style of **relative estimation followed by alignment** is widely used in the depth/geometry literature and is not specific to our work.
>
> > The observations did not provide key findings
>
> Our main contribution is to place MFMs and specialist models on exactly the same tasks and constraints, enabling direct, quantitative comparison. In particular, the benchmark reveals several key findings, as acknowledged by **Reviewers J3EM, 9qaY, and GMJw**:
>
> 1. MFMs are consistently behind specialists on all tasks, even after calibrating for the chaining constraints (“+chain”).
> 2. They are stronger on semantic tasks than on geometric ones.
> 3. The **ordering of MFMs** across tasks and the improvements from reasoning models on geometric tasks were not obvious a priori and emerge only from a controlled, cross-model evaluation.
>
> > Why the reasoning models work worse on semantic understanding?
>
> This appears to stem from a misreading of the results. Our experiments show that reasoning models (e.g., o1, o3) are **competitive or better** than non-reasoning models on most semantic tasks. For example, in **App. F.2**:
>
> - **Classification (Table 17):** o3 (**78.0%**) > GPT-4o (76.7%)
> - **Object Detection (Table 18):** o3 (**38.57 AP**) > GPT-4o (38.17 AP)
> - **Segmentation (Table 19):**  o3 (**38.18 mIoU**) > GPT-4o (35.66 mIoU)
>
> The only consistently weaker model is **o4-mini**, a lightweight distilled variant. We will clarify this distinction more prominently in the camera-ready.
>
> > metrics for surface normal and depth are not the most popular ones.
>
> For MFMs, we only obtain **ordinal** predictions, so it’s appropriate to choose metrics that are well-matched to this setting [4]. For depth, we use rank-based consistency (Spearman correlation) as the primary indicator of geometric quality, and report common error metrics (AbsRel, delta) **after** the standard scale-and-shift alignment [2]. We choose rank-based consistency for normals for the same reason.
>
> > Results with GPT-4o image generation is confusing and did not fit the whole story…
>
> We believe the image-generation experiments are a valuable addition as they address an obvious alternative strategy: *“Why not just ask the model to generate the depth/normal map?”* Our preliminary analysis (Fig. 5 & 9) shows that  GPT-4o tends to produce hallucinated structures, with frequent misalignments between the input and generated image. This suggests that, in its current form, native image generation is not yet accurate enough for precise quantitative benchmarking. In contrast, the text-based geometric querying we adopt yields tractable outputs. We will clarify this motivation in the text.
>
> > “How will random superpixelizations/grid seeds affect the performance?”
>
> We analyzed sensitivity to discretization in **App. E.8 (Fig. 22)**. While finer granularity improves **Oracle** and **Specialist** performance, MFM performance saturates quickly. Since the gap to specialists persists (and even widens) across variations, our conclusions are robust to specific superpixel or grid definitions.

---

> > ### Author Response · Authors · 2025-11-25
> > **Response to reviewer hd9G**
> >
> > > Could other CV tasks also be evaluated in the same manner, such as pose estimation, edge detection?
> >
> > Yes, in principle, our framework is generic: any task whose output can be discretized into a finite set of sub-tasks can be translated into chained textual queries. For example, edge detection could be handled via classifying superpixels or small patches as “edge” vs. “non-edge.” We will expand the discussion section to explicitly mention these extensions as directions for future work.
> >
> > [1] Depth Anything: Unleashing the Power of Large-Scale Unlabeled Data, Yang et al., 2024
> >
> > [2] Towards Robust Monocular Depth Estimation: Mixing Datasets for Zero-shot Cross-dataset Transfer, Ranftl et al., 2019
> >
> > [3] Learning Ordinal Relationships for Mid-Level Vision, Zoran et al., 2015
> >
> > [4] Monocular Depth Estimation Using Relative Depth Maps, Lee et al., 2019

---

> ### Comment · Reviewer_hd9G · 2025-11-26
> **Response to Authors**
>
> Thanks to the authors for the rebuttal. While the Oracle+Chain explanation is helpful, significant concerns regarding the validity and reliability of the framework remain. Some of my concerns have been resolved, however, some still remains:
>
> **Validity of Geometric Evaluation**:
> - The evaluation lacks transparency. No visual comparisons between *Specialist+Chain* and *Oracle+Chain* are provided, making it impossible to verify the correctness of the implementation.
> - Furthermore, the quantitative scores for these two baselines are suspiciously close, suggesting a ceiling effect in the chaining pipeline itself rather than accurately measuring the gap in MFM performance.
>
> **Redundancy of Semantic Tasks**: Without a reliable geometric evaluation, the benchmark essentially re-evaluates image classification under different cropping strategies. Object Detection, Segmentation, and Grouping are methodologically reduced to classification variants here. The high correlation in performance is expected, which limits the breadth of the contribution.
>
> **Generalization Claims (Edge Detection)**: I respectfully disagree that this framework generalizes to tasks like Edge Detection. Superpixels inherently group pixels, which would force edges to appear "thick" and "curvy" due to aliasing. These pipeline-induced artifacts make the task unsolvable for VLMs, similar to the issues observed in the geometric tasks.

---

> ### Author Response · Authors · 2025-12-01
> **Response to reviewer hd9G**
>
> Thank you for the follow-up and for engaging in the discussion. We address the remaining concerns below:
>
> > “the quantitative scores for these two baselines are suspiciously close, suggesting a ceiling effect”
>
> We show that this can be mitigated by extending the App. E.8 experiment for depth and normals. Concretely, we increase the number of superpixels (up to 2000) and pairwise comparisons. As shown in the tables below, both **Oracle+Chain** and **Specialist+Chain** scores improve with increasing granularity, and the gap between them **widens.** In such a discretization-and-reconstruction pipeline, one expects that as the granularity increases (i.e., more superpixels), the correlation obtained from the reconstructed map should approach the correlation of the native dense prediction; this is exactly what we observe, with both scores converging toward the non-superpixelated baselines.
>
> This behavior is consistent with the intended behavior of the pipeline: it successfully recovers finer-grained geometry (see below) when provided with finer-grained inputs, and the **"ceiling" can be raised in a straightforward way**.
>
> - Depth ($\rho$)
>
> |  | Dense | 100 | 200 | 400 | 1000 | 2000 |
> | --- | --- | --- | --- | --- | --- | --- |
> | Oracle | 1.000 | 0.938 | 0.958 | 0.964 | 0.978 | 0.982 |
> | Specialist | 0.969 | 0.924 | 0.943 | 0.952 | 0.958 | 0.962 |
>
> - Normals ($\rho_z$)
>
> |  | Dense | 100 | 200 | 400 | 1000 | 2000 |
> | --- | --- | --- | --- | --- | --- | --- |
> | Oracle | 1.0 | 0.743 | 0.774 | 0.802 | 0.831 | 0.848 |
> | Specialist | 0.805 | 0.713 | 0.735 | 0.751 | 0.762 | 0.769 |
>
> > “No visual comparisons between Specialist+Chain and Oracle+Chain are provided”
>
> Thank you for raising this point; we agree it is useful to show the raw outputs. We have attached visual comparisons for the experiment above and will include them in the camera-ready version. You can find the links to the visuals for depth and normals [here](https://imgur.com/a/IWPLTWb) and [here](https://imgur.com/a/Jj2aZMx).
>
> > "ceiling effect in the chaining pipeline itself rather than accurately measuring the gap in MFM performance...”
>
> For a fixed granularity $g$ (e.g., number of superpixels), the `Oracle+Chain` defines a performance ceiling, say $C(g)$. However, this ceiling is only problematic if the models being compared (e.g., a model with performance $A$ and another with performance $B$) **are both close to it** ($A \approx B \approx C)$.
>
> In our setting, **MFMs are well below this ceiling** ($A \ll C$), so the measurement of the gap **relative to the specialist** ($B$) is significant. For example, in depth prediction, the spearman correlations are:
>
> - MFM ($A$): 0.54
> - Specialist ($B$): 0.81
> - Ceiling ($C$): 0.83
>
> The relative gap is substantial: $\frac{|A-B|}{C} = 33$%. Since $A$ is **far from the ceiling**, the conclusion is that the underperformance of MFMs is driven by model limitations, and not the pipeline’s ceiling $C(g)$.
>
> App. E.8 further supports this: increasing the number of superpixels and pairwise comparisons **preserves the MFM-specialist gap.** The absolute value of the ceiling is therefore **not** what our conclusions hinge on; what matters is the much larger gap between MFMs and specialists.
>
> > Redundancy of Semantic Tasks
>
> - We respectfully disagree, as it can be reductionist to view each task as simple classification. Classifying a global image crop is different from querying for the presence of specific object parts at different scales, determining the background class in a superpixel, or resolving grouping relationships.
> - Model rankings do differ across tasks. For example, Claude 3.5 Sonnet outperforms o4-mini on Classification (62.9% vs 55.9%) but is significantly worse on Object Detection (31.7 $AP_{50}$ vs 42.9 $AP_{50}$).
> - These tasks are not interchangeable: they all have their own standard benchmarks, metrics, and specialist models. A key value of our benchmark is enabling direct comparison between MFMs and these task-specific specialists (e.g., OneFormer for segmentation or SAM for grouping) using standard metrics, which is impossible if we treat everything as global classification.
> - Casting these semantic tasks as classification-style queries is not specific to our work: it follows a long line of methods that treat detection and segmentation as per-region or per-pixel classification problems [1, 2]. In our case, this choice is a consequence of current MFMs being trained to output text, so the classification interface reflects the models’ native output format. For object detection, we further experimented with directly regressing the bounding box coordinates. As demonstrated by the results in **Tab. 9 (App. E. 1)**, direct regression does not work for all MFMs.

---

> ### Author Response · Authors · 2025-12-01
> **Response to reviewer hd9G**
>
> > Generalization Claims (Edge Detection)
>
> We appreciate the clarification here and agree that a naive use of superpixels might not work. Our intent was to argue that the general idea (discretizing outputs into text-queryable units) can be extended to other tasks, not that the current superpixel way of discretization is immediately suitable for edge detection as-is. We will add a clarifying note in the camera-ready.
>
> [1] Rich feature hierarchies for accurate object detection and semantic segmentation, Girshick et al., 2013
>
> [2] Class Segmentation and Object Localization with Superpixel Neighborhoods, Fulkerson et al., 2009

---

### Official Review · Reviewer_J3EM · 2025-11-01

**Soundness:** 2
**Presentation:** 3
**Contribution:** 2
**Rating:** 6
**Confidence:** 2

**Summary:**

This paper presents a comprehensive benchmark evaluating multimodal foundation models (MFMs) on standard computer vision tasks including classification, object detection, semantic segmentation, grouping, depth estimation, and surface normal prediction. The key contribution is a prompt chaining framework that decomposes complex vision tasks into text-solvable sub-tasks, enabling API-level evaluation of closed-source models. The study evaluates GPT-4o, o4-mini, Gemini 2.0 Flash, Gemini 1.5 Pro, Claude 3.5 Sonnet, Qwen2-VL, and Llama 3.2 on established datasets. Main findings show that: (1) MFMs lag significantly behind specialist models on all tasks, (2) they perform better on semantic tasks than geometric ones, (3) GPT-4o performs best among non-reasoning models, and (4) reasoning models (o1, o3, o4-mini) show promising improvements on geometric tasks.

**Strengths:**

- Extensive Model Coverage: The evaluation includes a diverse set of models (7 main MFMs plus reasoning models), providing a thorough landscape of current multimodal model capabilities across both open and closed-source systems.

- Multiple Task Evaluation: The breadth of tasks evaluated (6 core vision tasks) spanning semantic to geometric understanding provides valuable insights into where MFMs excel and struggle.

**Weaknesses:**

- Potential Data Contamination: While the authors conduct in-the-wild evaluations to address this, the use of standard benchmarks (ImageNet, COCO) raises concerns about training data leakage for closed-source models, which could inflate performance estimates.

- Limited Analysis of Failure Modes: While the paper shows that MFMs struggle with geometric tasks, there is limited investigation into why they fail (e.g., lack of 3D training data, architectural limitations, reasoning deficits). The "blurry vision" hypothesis is mentioned but not systematically explored.

- Task Selection Bias: The choice of tasks favors dense prediction problems that can be decomposed into classification. Other important vision capabilities (e.g., visual reasoning, fine-grained recognition, video understanding) are not evaluated.

**Questions:**

Please refer to the weaknesses.

---

> ### Author Response · Authors · 2025-11-25
> **Response to reviewer J3EM**
>
> We thank reviewer J3EM for their time and constructive feedback. We address the main concerns below.
>
> > "Potential Data Contamination: …"
>
> This is a valid concern [1]. We addressed it experimentally in Section 4.2 (“in-the-wild evaluations”) by testing on images collected after the models were released to minimize leakage. In this setting, we observe the same broad trends, supporting the robustness of our conclusions. We believe this constitutes a reasonable effort to mitigate contamination risks in the absence of training data or weights of these closed-source models.
>
> > Limited Analysis of Failure Modes
>
> We agree that understanding *why* MFMs fail on geometric tasks is important. While the closed nature of these models prevents us from directly probing training data or internal representations, we do analyze concrete failure patterns along several axes:
>
> 1. We find that simply outlining individual superpixels on the input image leads to a sharp drop in performance (Table 16), whereas providing multi-scale crops of each superpixel substantially improves results. This suggests that current MFMs struggle to resolve fine-grained details when presented with the full global context alone, which is consistent with prior findings that MFMs have “blurry vision” and struggle with fine-grained localization [2, 3].
> 2. Our **blind-guess depth** analysis (App. E.7) shows a systematic “ceiling bias”: models tend to assign larger depth to pixels higher in the image. We also observe this trend empirically across several images, where models consistently attribute greater depth to pixels higher in the image.
> 3. Third, in **surface normals**, we observe persistent coordinate-frame ambiguities, such as sign flips in the *x* and *y* components. We find that these errors can be partially resolved by CoT prompting (App. H).
>
> Taken together, these experiments offer a more systematic characterization of the failure modes. The results suggest that MFMs rely heavily on coarse semantic and layout cues while exhibiting unstable fine-grained geometric understanding. These observations are in agreement with other works that report a similar overall pattern, often described as a form of “blurry vision.” We thank the reviewer for their feedback and will add a clearer discussion on these in the camera-ready.
>
> > Task selection bias, missing visual reasoning / fine-grained / video.
>
> We agree that our benchmark does not cover the full spectrum of visual capabilities (e.g., video, fine-grained recognition, high-level visual reasoning). Our goal in this paper is more focused: we **systematically evaluate image-based semantic and geometric understanding on classic vision benchmarks** (classification, detection, segmentation, depth, normals) under a single framework. We see this as a **building block**. Extending evaluations to video and complex reasoning tasks is an important direction for future work, and we will add a discussion to explicitly clarify this scope.
>
> [1] Stop Uploading Test Data in Plain Text: Practical Strategies for Mitigating Data Contamination by Evaluation Benchmarks, Jacovi et al., 2023
>
> [2] BLINK: Multimodal Large Language Models Can See but Not Perceive, Fu et al., 2024
>
> [3] V*: Guided Visual Search as a Core Mechanism in Multimodal LLMs, Wu et al., 2023

---

### Author Response · Authors · 2025-12-03
**Response to All Reviewers**

We thank all reviewers for their time and constructive feedback. We are encouraged that they broadly agree on the value of the benchmark and framework.

All the reviewers explicitly acknowledge the **breadth** and **care** of the evaluation:

- Reviewer **J3EM** highlights the “extensive model coverage” and “multiple task evaluation… spanning semantic to geometric understanding” which provide “valuable insights into where MFMs excel and struggle”.
- Reviewer **hd9G** found the experimental evaluations “comprehensive”, and the prompt-chaining framework “interesting” and “effective for semantic tasks”.
- Reviewer **9qaY** describes our work as “one of the most exhaustive and systematic evaluations to date” and notes that the prompt-chaining framework is “thoughtfully designed” with “carefully calibrated” baselines.
- Reviewer **GMJw** finds our superpixel and grid-based prompting “a very clever way to prompt VLMs to perform dense prediction tasks” and emphasizes that the experiments are “extensive”.

### 1. Summary of Rebuttal & Clarifications

During the rebuttal, we addressed specific concerns and added new experiments. We summarize the main points of the discussion below:

- **Reviewer GMJw:** The reviewer noted that some baselines were not the absolute state-of-the-art. We have now added experiments with **Lotus**, **MoGe2**, and **DSINE** (for depth/normals) and **SAM2** (for grouping) under both their native protocols and our chaining setup. The results confirm that the gap persists, and that chained specialists substantially outperform MFMs.
- **Reviewer hd9G**: The reviewer inquired whether the discretization in the geometric pipeline imposes a performance ceiling that affects benchmarking validity. We clarified that our evaluation is explicitly **calibrated**: the benchmark hinges on the relative gap between MFMs and the `Specialist+Chain` baseline rather than absolute numbers. We showed that even **under identical chaining constraints, MFMs perform far below specialists**. Furthermore, we demonstrated experimentally that the constraints of chaining are mitigated by increasing superpixel granularity; this raises the pipeline's upper bound, while the gap between the MFMs and the specialists persists, confirming the significance of our conclusions.
- **Reviewer J3EM:** The reviewer raised concerns about potential training-data contamination on standard CV benchmarks. We acknowledged that contamination is an inherent challenge for closed-source MFMs and pointed to our “in-the-wild” evaluation on images collected **after** model release, where we observe the same trends. We could not find any sign suggesting this could be a concern, despite our search and control. To address the concern about failure-mode analysis, we elaborated on several experiments in the paper that probe how MFMs fail, including their limited ability to resolve fine-grained detail, consistent biases in depth, and recurring coordinate-frame ambiguities in surface normals.
- **Reviewer 9qaY:** The reviewer asked about distinguishing "translation error" (from the chaining) from actual model error. We clarified that our `Oracle+Chain` baseline is designed exactly for this purpose: it quantifies the drop caused by the chaining mechanism itself. Since MFMs perform significantly worse than the `Oracle+Chain` baseline, the remaining gap is attributable to the MFMs' lack of visual understanding.



### 2. References used in the rebuttal

###

[1] Stop Uploading Test Data in Plain Text: Practical Strategies for Mitigating Data Contamination by Evaluation Benchmarks, Jacovi et al., 2023

[2] BLINK: Multimodal Large Language Models Can See but Not Perceive, Fu et al., 2024

[3] V*: Guided Visual Search as a Core Mechanism in Multimodal LLMs, Wu et al., 2023

[4] Depth Anything: Unleashing the Power of Large-Scale Unlabeled Data, Yang et al., 2024

[5] Towards Robust Monocular Depth Estimation: Mixing Datasets for Zero-shot Cross-dataset Transfer, Ranftl et al., 2019

[6] Learning Ordinal Relationships for Mid-Level Vision, Zoran et al., 2015

[7] Monocular Depth Estimation Using Relative Depth Maps, Lee et al., 2019

---

### Meta-Review · Area_Chair_QpQp · 2026-01-07

**Summary:**

The submission proposes a method for evaluating multimodal foundation models on standard CV tasks including both semantic tasks and geometric tasks. The author study various models (GPT-4o, o4-mini, Gemini 2.0 Flash, Gemini 1.5 Pro, Claude 3.5 Sonnet, Qwen2-VL, and Llama 3.2) on various tasks (classification, object detection, semantic segmentation, grouping, depth estimation, and surface normal prediction). They construct different pipelines for different task, specifically, use pairwise ranking algorithm for geometric tasks like depth and normal estimation. The results exhibits MFMs perform better on semantic tasks than geometric tasks, and all MFMs lag significantly behind specialist models on all tasks.

Reviewers mainly concern on task choices, the availability on geometric task, superpixel size and analysis on the phenomena.

**Reviewer Concerns:**

Addressed Concerns:

Reviewer J3EM:
  1. Potential Data Contamination: Tested with images collected after the models release date, and observed the same phenomena.
  2. Limited Analysis of Failure Modes: More experiments are provided in probing how MFMs fail.

Reviewer hd9G:
  1. Numerical results cannot be gained with ranks: The evaluation metric is based on relative geometry, not reconstruction.
  2. No obvious key findings.
  3. Redundancy of Semantic Tasks: model performance and rankings are varying across tasks.

Reviewer 9qaY:
  1. Prompt Chaining Overhead and Realism: The prompt chaining is not suitable for deployment but the one-time evaluation makes the benchmark still accessible for the community.
  2.  Insufficient Error Analysis on Geometric Tasks: More experiments are provided in probing how MFMs fail.

Reviewer GMJw:
  1. Outdated specialist models: Additional experiments is added.
  2. Influence of Superpixel size in geometric tasks: Increasing the number of superpixels yields only marginal improvements.

Remaining Concerns:

Reviewer hd9G:
  1. Validity on the geometric tasks & Ceiling Effect.
  2. Generalization Claims: The designed superpixel pipeline is not suitable for tasks like pose estimation and edge detection.

Reviewer J3EM:
  1. Task selection bias: The method only cover classic CV tasks and lacks visual reasoning, fine-grained recognition, video understanding.

Reviewer 9qaY:
  1. Limited Exploration of Advanced Prompting or Visual Tools: Some variants of implementation are explored, but advanced prompting techniques are not explored.

**Reviewer Scores:**

Reviewer J3EM: rank 6, would be likely to remain the rank.

Reviewer hd9G: rank 2, would be likely to remain the rank for not agreeing on the geometric task design.

Reviewer 9qaY: rank 8, would be likely to remain the rank.

Reviewer GMJw: rank 8, would be likely to remain the rank.

The proposed method of evaluating MFMs on classic CV tasks is interesting and somehow reveals the deficiencies of understanding fine-grained contents for VLMs.

---

### Decision · Program_Chairs · 2026-01-26

Accept (Poster)